# Uncertainty of gridded precipitation and temperature reference datasets in climate change impact studies

Mostafa Tarek[1,2], François Brissette[1], and Richard Arsenault[1]

[1]Hydrology, Climate and Climate Change Laboratory, École de technologie supérieure, 1100 Notre-Dame West, Montreal, Quebec, Canada, H3C 1K3
[2]Department of Civil Engineering, Military Technical College, Egypt.

**Correspondence:** Mostafa Tarek (mostafa-tarek-gamaleldin.ibrahim.1@ens.etsmtl.ca)

**Abstract.** Climate change impact studies require a reference climatological dataset providing a baseline period to assess future changes and post-process climate model biases. High-resolution gridded precipitation and temperature datasets interpolated from weather stations are available in regions of high-density networks of weather stations, as is the case in most parts of Europe and the United States. In many of the world's regions, however, the low density of observational networks renders

gauge-based datasets highly uncertain. Satellite, reanalysis and merged products dataset have been used to overcome this deficiency. However, it is not known how much uncertainty the choice of a reference dataset may bring to impact studies. To tackle this issue, this study compares nine precipitation and two temperature datasets over 1145 African catchments to evaluate the dataset uncertainty contribution to the results of climate change studies. These deterministic datasets all cover a common 30-year period needed to define the reference period climate. The precipitation datasets include two gauged-only products

(GPCC, CPC Unified), two satellite products (CHIRPS and PERSIANN-CDR) corrected using ground-based observations, four reanalysis products (JRA55, NCEP-CFSR, ERA-I, and ERA5) and one gauged, satellite, and reanalysis merged product (MSWEP). The temperature datasets include one gauged-only (CPC Unified) product and one reanalysis (ERA5) product.

     All combinations of these precipitation and temperature datasets were used to assess changes in future streamflows. To assess dataset uncertainty against that of other sources of uncertainty, the climate change impact study used a top-down hydroclimatic

modeling chain using 10 CMIP5 (fifth Coupled Model Intercomparison Project) General Circulation Models (GCMs) under RCP8.5 and two lumped hydrological models (HMETS and GR4J) to generate future streamflows over the 2071-2100 period. Variance decomposition was performed to compare how much the different uncertainty sources contribute to actual uncertainty.

     Results show that all precipitation and temperature datasets provide good streamflow simulations over the reference period, but 4 precipitation datasets outperformed the others for most catchments: they are, in order: MSWEP, CHIRPS, PERSIANN,

and ERA5. For the present study, the 2-member ensemble of temperature datasets provided negligible levels of uncertainty. However, the ensemble of nine precipitation datasets provided uncertainty that was equal to or larger than that related to GCMs for most of the streamflow metrics and over most of the catchments. A selection of the best 4 performing reference datasets (credibility ensemble) significantly reduced the uncertainty attributed to precipitation for most metrics, but still remained the main source of uncertainty for some streamflow metrics. The choice of a reference dataset can therefore be critical to climate

change impact studies as apparently small differences between datasets over a common reference period can propagate to generate large amounts of uncertainty in future climate streamflows.

## 1 Introduction

General Circulation Models/Earth System models (ESM) /Global Climate Models (GCMs) are the primary tools used to simulate the response of the global climate system to increases in greenhouse gas concentrations and to generate future climate projections. GCMs are complex mathematical representations of the physical and dynamical processes governing atmospheric and oceanic circulations as well as the interactions with the land surface. In order to reduce the computation burden, which can be considerable, GCMs represent the earth with a grid having a relatively coarse spatial resolution (IPCC, 2001). Consequently, GCM projections cannot be used directly for fine scale climate impact studies. Statistical/empirical or dynamical downscaling techniques have thus commonly been used to address this scale mismatch. In addition, climate model outputs are always biased, and the extent of these biases can be evaluated through a comparison against observations over a common reference period. A bias correction procedure is therefore generally performed in addition to the downscaling step, and biases are assumed to be invariant in time when the correction is applied to future climate projections (Velázquez et al., 2015). Although a two-step downscaling bias correction approach is preferable in most cases, a single instance of bias correction is sometimes used to account for both scale mismatch and GCM biases. While this may be acceptable when the scale difference is small (e.g., when using catchment averaged values), recent studies have shown that bias correction has limited downscaling skills (Maraun, 2016).

Statistical downscaling and bias correction approaches primarily rely on hydrometeorological observations over a historical reference period. It is therefore primordially important that the observed reference dataset represents the true climate state as closely as possible. For this task, ground stations remain the standard and most accurate/trusted source of weather data (New et al., 2001; Nicholson, 2013). However, the spatial distribution of these stations varies widely across the globe, and coverage is often sparse and even deficient in many parts of the world outside of Europe and the US. Even in well-covered regions, gauge data is subject to many problems, such as missing data, precipitation undercatch and inhomogeneities related to a variety of issues such as equipment change, station relocation and land surface modifications near each station (Kidd et al., 2017; Peterson et al., 1998).

In recent decades, extensive efforts have been devoted to the development and improvement of gridded global and quasi-global climate datasets to overcome the limitations of gauge stations. These datasets provide meteorological record time series with continuous spatiotemporal coverage, and typically, no missing data. However, various error sources are inherent in these datasets, thus also bringing uncertainty to the data (Voisin et al., 2008). Thus, choosing an appropriate reference dataset for climate change impact studies is an important concern, and especially so in regions with sparse ground station coverage.

According to Huth (2004): "For estimates based on downscaling of General Circulation Model (GCM) outputs, different levels of uncertainty are related to: (1) GCM uncertainty or inter-model variability, (2) scenario uncertainty or inter-scenario variability, (3) different realizations of a given GCM due to parameter uncertainty (inter-model variability) and (4) uncer-

tainty due to downscaling methods". In most climate change impact studies, it is generally assumed that GCMs are the major source of uncertainty (Mpelasoka and Chiew, 2009; Kay et al., 2009; Vetter et al., 2017). Rowell (2006) compared the effect of different sources of uncertainty using the initial condition ensembles of different GCMs, Greenhouse Gases Emission Scenarios (GHGES) and Regional Circulation Models (RCMs) on changes in seasonal precipitation and temperature in the United Kingdom. The results indicated that the largest uncertainty comes from the GCM choice. Minville et al. (2008) used ten equally-weighted climate projections derived from a combination of five GCMs, two GHGES and a single downscaling method for downscaling to investigate the uncertainty envelope of future hydrologic variables. Their results showed that the uncertainty related to the GCM choice is dominant. These results have also been confirmed by several studies (Prudhomme and Davies, 2009; Nóbrega et al., 2011; Dobler et al., 2012). Other studies have assessed other sources of uncertainty such as Greenhouse Gases Emission Scenarios (GHGES) (Prudhomme et al., 2003; Kay et al., 2009; Chen et al., 2011), the downscaling method (Wilby and Harris, 2006; Khan et al., 2006) and hydrological modeling (Bae et al., 2011; Vetter et al., 2017). Recent studies have also looked at the uncertainty related to the choice of the impact model (Giuntoli et al., 2018; Krysanova et al., 2018). From these studies, a more complex picture emerges, in which the main source of uncertainty may vary, depending on geographical location and metric under study. Dataset uncertainty has been assessed in numerous studies either by direct inter-comparison between datasets (Vila et al., 2009; Andermann et al., 2011; Romilly et al., 2011; Jiang et al., 2012; Chen et al., 2014; Prakash et al., 2018; Nashwan and Shahid, 2019) or by using hydrological modeling (Behrangi et al., 2011; Beck et al., 2017b; Wu et al., 2018; Zhu et al., 2018; Tarek et al., 2019). However, to the best of our knowledge, the uncertainty of gridded datasets has not been evaluated against other sources of uncertainties when performing climate change impact studies. The objective of this study is therefore to assess the impact of the choice of a given reference dataset on the global uncertainty chain of climate change impact studies. Since this is of particular concern to regions with sparse weather station coverage, this study is conducted over Africa.

## 2   Study Region and data

### 2.1   Study Region

#### 2.1.1   Geographic situation

Africa is the second largest and second most-populous continent in the world. It covers a land area of about 30.3 million km$^2$, including adjacent islands, which represents 6% of Earth's total surface area and 20.4% of its total land area (Mawere, 2017). Deserts and dry lands cover 60% of its entire surface (Prăvălie, 2016). The average elevation of Africa is almost 600 m above sea level, roughly close to the average elevations of North and South America (Atrax, 2016). Generally, higher-elevation areas lie to the east and south, while a progressive decrease in altitude towards the north and west is apparent.

The African continent can be divided into 25 major hydrological basins (Karamage et al., 2018). Generally speaking, the main drainage for all of the continent's basins is towards the north and west, and ultimately, into the Atlantic Ocean. About 95% of its streams are drained through permanent rivers. In some arid areas (i.e., Northwest Sahara Desert), drainage is sometimes

absent or masked by sand seas (Karamage et al., 2018). Roughly, 60% of the African continent is drained by 10 large rivers (Congo, Limpopo, Niger, Nile, Ogooue, Orange, Senegal, Shebelle, Volta and Zambezi) and their tributaries (Paul et al., 2014).

### 2.1.2 Climate profile

Africa is the hottest continent on earth, and is the area that has seen the highest ever recorded land surface temperature (58 °C in Libya; El Fadli et al. (2013)). The continent is characterized by highly variable climates that range from tropical to subarctic on its highest peaks. According to the Koppen climate classification (Köppen, 1900), the northern half is mainly classified as dry (group B) whereas the central and southern areas contain both savannah plains and dense forests with tropical and humid subtropical climates (groups A and C) with a semi-arid climate in-between (El Fadli et al., 2013). These wide climate ranges are characterized by a wide variety of precipitation extremes, including droughts and floods. Droughts occur mostly in the Sahel and in some parts of Southern Africa, whereas flooding is most prevalent in the southern and eastern regions. Looking at the more recent hydrological climate classification of Knoben et al. (2018), Africa can be classified as a no-snow continent, with a strong precipitation seasonality between the tropics and a high aridity index in the extratropical zones, as well as along the coast of the Indian Ocean in the tropical band.

## 2.2 Data

This project used several datasets built from climate models, observed precipitation, temperature and streamflow, as well as catchment boundaries. These are described in the following four sub-sections.

### 2.2.1 General Circulation Models (GCMs)

All GCMs used in this study were part of the Coupled Model Intercomparison Project Phase 5 (CMIP5) (Taylor et al., 2012). Long historical climate simulations (1850–2005) and future climate projections (up to 2100 and beyond) for four Representative Concentration Pathways (RCPs) are included in the CMIP5 database.

Ten CMIP5 GCMs from 10 different modeling centers were selected for this study, as shown in Table 1. They were selected as a subset of the GCMs used to set up the North American Climate Change and Hydroclimatology (NAC$^2$H) database (Arsenault et al., 2020). The number of GCMs (10) was selected as a compromise between having an accurate representation of GCM climate sensitivity variability and keeping the large computational burden of this project reasonable. All GCM data was extracted over the 1983-2012 and 2071-2100 future periods under the (RCP8.5) emission scenario.

### 2.2.2 Gridded precipitation and temperature datasets

The precipitation and temperature dataset selection was made on the basis of a high spatial resolution, daily (or higher) temporal resolution, and of the availability of at least 30 years of data covering the same time period, in order to properly define the reference climate. Some recent datasets that provide global/near-global rainfall information at finer spatial and temporal resolutions (e.g. the GPM Integrated Multisatellite Retrievals (IMERG) (Huffman et al., 2015), and the Global Satellite Map-

ping of Precipitation (GSMaP) (Okamoto et al., 2005) were left out because their temporal coverage was too short to properly represent the mean climate over the reference period.

According to above criteria, nine precipitation and two temperature datasets were selected for this study. The precipitation datasets include two gauged-only products, two satellite products corrected using ground-based observations, four reanalysis products and one gauge, satellite, and reanalysis merged product. The temperature datasets include one gauged-only and one
reanalysis product as shown in Table 2.

### 2.2.3    Observed streamflow data

The observed streamflow records were obtained from the Global Runoff Data Centre (GRDC) archive. The GRDC is arguably the most complete global discharge database providing free access to river discharge data (Fekete and Vörösmarty, 2007). The database provides streamflow records collected from 9213 stations across the globe, with an average temporal coverage
of 42 years per station (Do et al., 2017). It is operated under the World Meteorological Organization (WMO) umbrella to provide broad hydrological data to support the scientific research community. GRDC data has been widely used in various hydrological studies, such as those examining hydrological model calibrations (Milliman et al., 2008; Hunger and Döll, 2008; Donnelly et al., 2010; Haddeland et al., 2011), or as a benchmark to compare simulated streamflows (Trambauer et al., 2013; Zhao et al., 2017). The streamflow records in the GRDC database have all undergone a quality control process, but there
is always the possibility that some level of regulation may affect the data (Tramblay et al., 2020). No direct homogeneity testing was performed to detect potential changes due to regulation, but an indirect quality assessment was done through the hydrological modeling performance during the calibration process.

### 2.2.4    Watersheds boundaries data

HydroSHEDS (the Hydrological data and maps based on the SHuttle Elevation Derivatives at multiple Scales database) is a
freely available global archive, developed through a World Wildlife Fund (WWF) program, that uses a hydrologically-corrected digital elevation model to provide hydrographic information for regional and global studies (Lehner et al., 2008). In addition, it applies a consistent methodology using Geographic Information System (GIS) technology to provide watershed polygons for more than 7000 GRDC gauging stations. Figure 1 shows watershed polygon layers at different spatial scales for the African continent. The vector layer (Lev05), which consists of 1145 watersheds, was chosen to be used in this study. It was selected
as a compromise between having a sensible number of watersheds and keeping the large computational burden of this project reasonable.

## 3    Methodology

Figure 2 presents the methodological framework for this study. A large-sample hydrological climate change impact study is performed over 1145 African catchments. It uses the standard top-down approach in a modeling chain, which consists of 10
GCMs, 2 hydrological models, 2 temperature and 9 precipitation datasets, for a total of 360 possible combinations. A single

GHGES (RCP8.5), a single climate projection for each GCM and a single downscaling method (see below) are used, since the focus of this work is not on conducting a complete uncertainty chain study. The uncertainty related to the reference dataset will therefore be compared to that of the climate model ensemble and against that of both hydrological models. These two sources are generally considered to be the most important in climate change impact studies (e.g. Giuntoli et al., 2018; Krysanova et al., 2018). For each catchment, 360 30-year streamflow time series are generated for both the reference (1983-2012) and future (2071-2100) time periods. Six streamflow metrics are computed for each of these time series. An n-dimensional analysis of variance is performed to partition the uncertainty linked to the four selected groups of components of the uncertainty modeling chain: precipitation and temperature datasets, GCMs and hydrological models.

Both hydrological models were calibrated on all catchments for all 18 combinations of reference datasets (2 temperature datasets x 9 precipitation datasets), for a total of 41,220 independent hydrological model calibrations. Combining different - and somewhat independent - data sources for temperature and precipitation raises potential issues about mass and energy balance. Most of the products used in this work originate from a gridding process that is independently done for precipitation and temperature, therefore not taking into account temporal correlations between both variables. Most precipitation products are also developed independently of temperature. Reanalyses are the most consistent dataset with respect to energy budget and water balance. However, even though the weather model of the reanalysis is entirely physically coherent, the data assimilation does not preserve this physical coherency and therefore reanalysis does not conserve water balance. The combination of precipitation and temperature datasets is therefore unlikely to be problematic. More details about the calibration process are described later in section 3.1.3.

The watershed boundaries for the African continent were extracted from the HydroSHEDS database. Streamflow records from the GRDC database were used to calibrate the hydrological models and to evaluate the hydrological modeling performance. In this study, 350 stations were chosen from the GRDC database based on three criteria. First, stations should have data for the 1983-2012 study period. Second, stations that have less than five consecutive years of data during this period were excluded. Finally, all the stations should be compatible with the selected HydroSHEDS catchments. In order to include additional catchments to allow for a better spatial coverage over the African continent, an additional 795 catchments (the remaining catchments from the Lev05 layer of Figure 1) were selected and an additional regionalization step was performed to generate streamflows at these 795 catchments. The climatological data from 9 precipitation and 2 temperature datasets were then extracted for each of these 1145 catchments. The main methodological steps are described below in Figure 2.

## 3.1 Hydrological modeling

Given the large-scale nature of this study, distributed and physically-based models were not considered. Two lumped hydrological models, GR4J and HMETS, were selected and calibrated over each of the 350 gauged catchments. The two hydrological models have been shown to perform well in a wide range of studies and over a wide range of climate zones (Arsenault et al., 2018; Essou and Brissette, 2013; Gosset et al., 2013; Martel et al., 2017; Simonneaux et al., 2008; Tarek et al., 2019, 2020a; Valéry et al., 2014).

### 3.1.1 The GR4J hydrological model

The GR4J (Génie Rural à 4 paramètres Journalier) model is a four-parameter lumped and conceptual rainfall-runoff model (Perrin et al., 2003). This model has shown overall good performance in several studies across the globe (Aubert et al., 2003; Raimonet et al., 2018; Riboust et al., 2019; Westra et al., 2014; Youssef et al., 2018). The model requires daily precipitation, temperature and potential evapotranspiration (PET) as inputs to simulate the streamflow. The Oudin formulation (Oudin et al., 2005) was used in the present study to compute the daily PET series as it was shown to be simple and efficient.

### 3.1.2 The HMETS hydrological model

The HMETS hydrological model (Hydrological Model – École de technologie supérieure; Martel et al. (2017)) is more complex than GR4J, with 21 model parameters. It has four reservoirs (surface runoff, subsurface flow from the vadose zone reservoir, delayed runoff from infiltration and groundwater flow from the phreatic zone reservoir). HMETS uses the same Oudin PET formulation, but with scaling parameters to control the mass balance.

### 3.1.3 Hydrological model calibration

The nine precipitation and two temperature datasets were combined in their eighteen possible arrangements for analysis purposes. Due to the large number of calibrations to be performed (41,220 model calibrations), an automatic model parameter calibration approach was selected. The Covariance Matrix Adaptation Evolution Strategy (CMAES) algorithm was chosen because of its flexibility and robustness (Hansen et al., 2003). CMAES has been shown to be one of the best and fastest automatic calibration algorithms available (Arsenault et al., 2014; Yu et al., 2013).

All 30 years were used for calibration, and no validation step was performed following the work of Arsenault et al. (2018). They showed that validation and calibration skills are not necessarily correlated, and that adding more years to the calibration dataset improves the hydrological model performance and robustness. The Arsenault et al. (2018) study was performed on catchments which showed no signs of non-stationarity. We applied the same methodology here despite foregoing any testing for homogeneity. For regionalization purposes, the maximum parameter identifiability was deemed preferable and using a longer time period maximized the likelihood of parameter identifiability. The same also holds for simulation, in that in the absence of any knowledge a priori of the impacts of climate change, using the entire parameter set is prudent as it protects against highly variable changing conditions in the future. The calibration objective function was the Kling-Gupta efficiency (KGE) metric, introduced by Gupta et al. (2009) and modified by Kling et al. (2012). It is defined as a combination of equally-weighted bias, variance and correlation aggregate metrics. The KGE values theoretically range from negative infinity, implying an extremely poor performance of the model, all the way to one, suggesting a perfect performance. Pechlivanidis and Arheimer (2015) divided the KGE values into three performance groups: Bad (KGE < 0.4), Acceptable ($0.4 \leq$ KGE < 0.7) and Good (KGE $\geq 0.7$).

## 3.2    Regionalization

The transfer of hydrological information (i.e., model parameters or streamflow) from one catchment (gauged) to another (ungauged) is known as "Regionalization" (Razavi and Coulibaly, 2013). Regionalization can be conducted using two methods: 1) rainfall-runoff models/model-dependent method, which typically transfers the model parameters from one or more gauged watersheds to an ungauged watershed, and 2) hydrological model-independent methods, which transfer the streamflow directly from gauged to ungauged watersheds (Razavi and Coulibaly, 2013). In this paper, the model-dependent method was applied as

it has been used in several studies and has shown acceptable results (Merz and Blöschl, 2004; McIntyre et al., 2005; Boughton and Chiew, 2007; Cutore et al., 2007; Samaniego et al., 2010; Beck et al., 2016; Arsenault and Brissette, 2014; Saadi et al., 2019).

The three approaches, namely, the spatial proximity (S.P), physical similarity (P.S) and multi-linear regression (MLR) methods (Oudin et al., 2008), have been used to estimate the model parameters in ungauged catchments. First, the three approaches

were tested to find the best method to apply. Then, the best-performing precipitation-temperature datasets combination were used to feed the hydrological models and simulate the streamflow of the ungauged catchments. Based on the hydrological modeling performance on the 350 gauged catchments, as represented by the KGE calibration score, the MSWEP precipitation and ERA5 temperature datasets were found to be the best combination used in computing the streamflow for the 795 ungauged catchments. This regionalization study is one of the very few performed over Africa and will be detailed in another paper. It

showed that the best regionalization methods were consistent with the ones identified in other regions of the world, and that regionalization performance was similar to that obtained in studies elsewhere around the world.

## 3.3    Bias correction

Most climate change impact studies have been applying univariate bias correction methods to correct climate model outputs. Univariate approaches cannot account for the temporal dependence between precipitation and temperature (and other vari-

ables). For example, if a model has a cold temperature bias and a dry precipitation bias, these biases would be corrected individually, whereas in reality precipitation and temperature are correlated (e.g. Wu et al., 2013). Multivariate techniques have been introduced as an alternative to overcome this deficiency. In this study, the N-dimensional multivariate bias correction algorithm (MBCn) was used (Cannon, 2018).

MBCn is an image processing technique extension that transfers all statistical characteristics between the historical and

projected periods while preserving the change projected for all quantiles of the distribution. The algorithm consists of three main steps: (1) application of an orthogonal rotation to both model and observational data; (2) correction of the marginal distributions of the rotated model data using quantile mapping, and (3) application of an inverse rotation to the results. These three steps are repeated until the model distribution matches the observational distribution. This computational complexity is one disadvantage of that method, as it requires several iterations to correct the projected outputs. However, MBCn is arguably

the best-performing quantile-based method available (Adeyeri et al., 2020; Meyer et al., 2019).

### 3.4 Variance analysis

An n-dimensional analysis of variance (ANOVA-N) was used to quantify the contribution of the different uncertainty sources to the overall variance (Von Storch and Zwiers, 2001). This method has been applied in many previous studies for this purpose (Addor et al., 2014; Bosshard et al., 2013; Trudel et al., 2017). For each catchment, 360 values for each metric are obtained, each related to a unique combination of 1 GCM, 1 precipitation dataset, 1 hydrology model and 1 temperature dataset. The variance analysis attributes the percentage of the total variance of this vector of 360 values, separating the main effects (the independent contribution of each of the 4 components, and the interactions between these components. The interactions imply that the behaviour of one source depends on another source (for example, precipitation dataset may generate lots of variance with some GCM but not for others). Computing the main effect and first order interactions is relatively cheap, computationally speaking, but higher orders (which typically carry much less variance) become exponentially costlier. For the four uncertainty components under study (GCMs, precipitation and temperature datasets, and hydrological models), a total of 11 variance components can therefore be computed: 4 main effect components as well as 6 first-order, 3 second-order, and 1 third-order interaction components.

The analysis of variance was performed for six streamflow metrics out of the fifty-one metrics defined in Arsenault et al. (2020) for each of the 1145 catchments. These six metrics cover a wide range of streamflow conditions: mean annual (Mean Q), seasonal (Winter Q and Summer Q) values, the $5^{th}$ and $95^{th}$ distribution quantiles (QQ5 and QQ95, respectively), as well as annual daily extreme (QX1) metrics.

### 4  Results

This section outlines the main findings of the work. Figure 3 presents the calibration results for both hydrological models using all possible combinations of the 9 precipitation and 2 temperature datasets. Each boxplot consists of 350 KGE values corresponding to the calibration result for each of the 350 selected gauged catchments. Each box extends from the $25^{th}$ quantile to the $75^{th}$ quantile, with the median displayed as the red line within that range. The top and bottom whiskers (where shown) represent highest and lowest values. Red crosses are considered statistical outliers.

Results show that both hydrological models perform well, but that there are important differences between datasets. HMETS performs better than GR4J, with respective overall mean KGEs of 0.58 and 0.41. All the precipitation and temperature datasets result in acceptable median KGE simulations Pechlivanidis and Arheimer (2015).

Based on the models calibration performance, both temperature datasets perform very similarly across all combinations, with ERA5 generally slightly outperforming CPC. Figure 3 clearly shows that most of the variability seen originates from the precipitation datasets. Four precipitation datasets are ahead of the field. They are, in order of performance: the merged product MSWEP, followed by the two satellite datasets; CHIRPS and PERSIANN, and the ERA5 reanalysis dataset. The gauge-based precipitation datasets (e.g., GPCC and CPC), and the ERA-I reanalysis follow with a similar performance. Finally, the CFSR and JRA55 reanalysis are the worst-performing products for hydrological model calibration.

Table 3 presents the main results of the analysis of variance for the 2071-2100 period for the gauged catchments. It shows the relative variance for all main effect and first-order interactions of the four components of uncertainty under study, and for 6 streamflow metrics. The variance originating from second- and third-order interactions are summed up and presented in the last row. Results show that most of the variance consistently comes from 5 sources, for all 6 streamflow metrics. They are: precipitation datasets (P), GCMs, hydrological models (HM), interactions between precipitation datasets and GCMs (P-GCM), as well as interactions between precipitation datasets and hydrological models (P-HM).

Table 3 indicates that both the precipitation datasets and GCMs are the main contributors to variance, including through interactions (P-GCM). The hydrology models also generate some uncertainty, and in particular, through interaction with the precipitation datasets. All metrics exhibit a similar pattern, with the exception of the low-flow metric (QQ5), where precipitation, hydrological models and their interaction components (P-HM) are dominant, and for which GCM uncertainty is minimal. In almost all cases, the five highlighted components represent approximately 85% of the total variance. The average amount of variance introduced by both temperature datasets is less than 0.25% for all 6 different streamflow metrics.

To show cross-catchment variability, Figure 4 shows boxplots of the relative variance attribution results for the 5 main contributors to variance, as identified in Table 3, and for the same 6 streamflow metrics. The results are also decomposed into three parts: all 1145 catchments (A), as well as the 350 gauged (G) and 795 ungauged (U) catchments, in order to ensure that the regionalization process does not introduce undesirable effects on the results.

Figure 4 shows that the response of the gauged and ungauged catchments is very similar across all variance components and streamflow metrics, and that no major variance artifact is introduced by the regionalization step. Consequently, all further results will only be shown for all 1145 catchments, with no differentiation made between the gauged and ungauged ones.

The results show that there is considerable across-catchment variability, as shown by the extent of the boxplots, with GCM and P-GCM interaction being the most important, and most variable contributors to variance. As was shown in Table 3, the low-flow metric displays a pattern that is much different from the other five metrics, with HM being important and GCM, being the lowest. Also, the HM and P-HM show significant contribution to the uncertainty in the Summer Q metric. There is a relatively large difference between the two metrics representing high flows (QQ95 and QX1). While GCM dominates the former, a much larger part of the uncertainty is transferred to the precipitation dataset (P and P-GCM) for the latter.

In order to study the impact of spatial variability, Figure 5 presents the spatial distribution of the relative variance attribution for the five main contributors to variance of Table 3 and all 6 streamflow metrics. Mean Q, Winter Q, QX1 and QQ95 display somewhat similar spatial patterns. Summer Q and QQ5 metrics display somewhat similar spatial patterns. The largest precipitation uncertainty (P and P-GCM interactions) is found in the northern parts of Sub-Saharan Africa, between 0 and 30 °N. GCM uncertainty appears to be larger all around the coastlines of Africa. Hydrological model uncertainty is strongest for QQ5, but spatial patterns are fairly consistent across all 6 streamflow metrics. GCM uncertainty is strongly different for both Summer Q and Winter Q, likely because of the monsoon pattern. Above 20°N, there is generally less than 100 mm of total annual precipitation, and some level of care should therefore be taken when analyzing results in relative contribution to variance. The relative contribution to variance is not related to absolute mean streamflow values and therefore the colour scale

is the same for major rivers and smaller intermittent streams. Many of the catchments above 20°N run dry for a large part of the year.

In other words, a variance analysis of a metric with very little absolute variance could be misleading. Consequently, Figure 6 displays the standard deviation of the 360 streamflow values computed for each streamflow metric and for each watershed. Therefore, Figure 6 does not represent the variance contribution of any given component of the hydroclimatic chain, but represents the total variance of all components combined. A low value indicates that a streamflow metric shows little variability across its 360 values. This would be expected for example for catchments with a high aridity index resulting in very transient flow. The streamflow value for each metric is normalized per unit area to allow for a comparison of large and small watersheds in the same figure. Not surprisingly, the results demonstrate a larger variance along the equatorial band, where precipitation is largest. This pattern is particularly clear for the QQ95 high-flow metric. The catchment database is, however, large enough to show some catchments which exhibit a large variance, even in arid regions above 20°N and below 20°S.

Since some precipitation datasets are clearly better than others based on the hydrological model calibration results, it may not be entirely fair to compare precipitation uncertainty to GCM uncertainty. To investigate this further, the uncertainty contribution obtained when using all 9 precipitation datasets is compared to that of 3 sub-ensembles, as presented in Table 4. While ensemble 4 is composed of the clearly best-performing datasets for model calibration, the main goal here is to investigate the impact of dataset selection, not the definition of a credibility ensemble, as will be further discussed later.

Figure 7 presents the boxplots of percentages of variance for each catchment, for the five main contributors to variance for all 4 precipitation dataset ensembles of Table 4. Unsurprisingly, it shows that reducing the size of the precipitation ensemble results in a consistent decrease in the variance attributed to precipitation. Most of this reduction in variance comes from the P-GCM interaction term, although there is also a noticeable decrease in the main effect P component. The lost precipitation variance is transferred mostly to GCMs, and to a lesser extent, to hydrological modeling. The exception is the low-flow QQ5, where most of the variance is transferred to HM. Most of the drop observed is obtained by dropping the five worst precipitation datasets (ensemble 4), as no significant difference is observed between precipitation ensembles 3 and 4. Even in a reduced ensemble, precipitation datasets still provide between 10 to 20% of median variance, and more than 30% for the low-flow metric (QQ5) when taking into account the main effect and first-order interaction term.

Figure 8 presents the spatial distribution of the relative variance attribution for each of the 6 streamflow metrics after including only the four best overall precipitation datasets (Ensemble 4 of Table 4). This is the same as Figure 5, but with a reduced precipitation ensemble. Results outline that GCM uncertainty is the dominant source of uncertainty when using the reduced precipitation ensemble, with the exception of the low-flow metric, for which hydrological model uncertainty is dominant. There are, however, significant interactions between GCM and precipitation for all metrics, especially in the northern half of the continent. Otherwise, the observed spatial patterns are similar to the ones presented in Figure 5.

## 5 Discussion

Defining a reference climate dataset is an important but difficult task. A reference climate dataset is used as a benchmark for monitoring environmental changes and correcting climate model biases of future climate projections to assess future impacts of a changing climate. Data from weather stations is still mostly considered to be the most accurate representation of the current climate, despite suffering from several important issues, such as precipitation undercatch and inhomogeneities (Peterson et al., 1998). To allow for regular data coverage and remove missing data, it is now common practice to interpolate station data onto a regular grid. Such gridded datasets greatly simplify the processing of meteorological data for environmental studies at the regional, continental and global scales. However, even in regions with a good weather station coverage, gridded datasets using the same underlying data differ due to the different interpolation methods (Essou et al., 2016), and typically see an increase in the number of wet days and a decrease in the frequency of extreme events (Ensor and Robeson, 2008). In regions with scarce weather station coverage (such as Africa), interpolation becomes extrapolation, and is therefore potentially highly unreliable. In such cases, environmental studies have had to rely on additional sources of data, such as satellite and atmospheric reanalysis for environmental studies. Several inter-comparison studies have been done (e.g., Beck et al. (2017b); Essou et al. (2017)), including over Africa (Satgé et al., 2020; Dembélé et al., 2020). These studies outline a complex picture in which performance depends on scale, climate and data source, and for which no dataset consistently outperforms all of the others. Because of this, in data-sparse regions such as Africa, there is not only no commonly agreed upon reference dataset, but even no agreement on the optimal source of climate data (e.g., satellite vs. reanalysis), and different environmental studies have used completely different datasets. This is particularly problematic for climate change impact studies since there is no knowledge on how dataset uncertainty may propagate in the typical hydroclimatic modeling chain. The results presented in this study attempt to answer this question by comparing dataset uncertainty to other sources of uncertainty, such as that derived from GCMs.

Results show that most of the dataset uncertainty originates from precipitation. Temperature displays much smaller spatial and temporal variability than precipitation, and can therefore be a lot more reliably interpolated using the adiabatic lapse rate to account for elevation and terrain orientation in mountain areas. Precipitation interpolation is a much more challenging problem, which explains why most dataset intercomparison work has focused on this variable. Based on KGE performance over a common reference period, all nine precipitation datasets performed adequately in terms of hydrological modeling performance, but some clearly performed much better than others. This is in agreement with the results of Beck et al. (2017b) and Beck et al. (2019). The uncertainty contribution of datasets to future streamflow uncertainty was first evaluated using all 9 precipitation datasets, in conjunction with 2 temperature datasets, a sample of 10 GCMs and two hydrological models, for a total of 360 possible element combinations. While this is a relatively large sample, not all sources of uncertainty were accounted for. In particular, GHGES, downscaling and bias correction were not included in the analysis. In comparison, the North American Climate Change and Hydroclimatology Dataset (NAC[2]H) database (Arsenault et al., 2020) offers 16,000 combinations allowing examining future streamflow uncertainty. In this regard, the relative variance contribution of the climate dataset is best examined in comparison to that of GCMs, the most studied source of climate change impact uncertainty. Results outline the important, and in some cases, dominant contribution of the precipitation dataset to the overall uncertainty of future streamflows.

For all 6 streamflow metrics presented here, the precipitation dataset uncertainty was comparable and sometimes larger than that of GCMs.

Uncertainty contribution was then studied by retaining subsets of precipitation datasets, eliminating the least performing ones with respect to the chosen KGE metric. This follows the concept of a credibility ensemble based on carefully selecting the best/most robust components of the hydroclimatic modeling chain, in order to obtain the most credible uncertainty range ((Brissette et al., 2020; Giuntoli et al., 2018; Maraun et al., 2017)). Results demonstrate a large decrease in contribution to uncertainty for 5 of 6 streamflow metrics. The precipitation dataset remained the largest contributor to uncertainty for the low-flow metric, and still accounted for 10 to 20% of the total variance for the other metrics. Most of the decrease in uncertainty was obtained by dropping the worst-performing datasets, rather than keeping the best-performing ones.

The results presented here indicate that hydrological model uncertainty is relatively small, with the exception of the low-flow metric. These results should be taken with caution because only two hydrological models were used, and also because they both share the same potential evapotranspiration (PET) formula. For climate change impact studies, the climate sensitivity of PET is now thought to be an important source of uncertainty for impact studies (Clark et al., 2016; Brissette et al., 2020), and the importance of hydrological model uncertainty has been outlined in many studies (Vetter et al., 2017; Krysanova et al., 2018; Giuntoli et al., 2018). A better understanding of how hydrological model components affect uncertainty would therefore be very valuable for climate change impact studies (e.g. Dallaire et al., 2021; Duethmann et al., 2020; van Kempen et al., 2021). Taking the above into consideration, it is therefore likely that the contribution of hydrological models is underestimated here. The number of components in a variance attribution study is an important issue. However, the contribution to variance is related to how dissimilar the ensemble members are and not strictly to its numbers. As such, the ensemble with the fewest members can still provide the largest contribution to variance. There is therefore no need for all ensembles to have the same number of members, but rather to have enough credible members to cover the uncertainty. Despite having only two temperature datasets here, adding more temperature datasets is unlikely to change the results considering how little uncertainty is present in the two datasets when compared to other sources. Temperature is the easiest variable to measure and to extrapolate, especially when compared to precipitation. It is therefore expected that precipitation uncertainty would normally dwarf the contribution of temperature. Based on previously published work, 10 GCMs is very likely more than enough to frame the uncertainty contribution from this source (e.g. Wang et al., 2020).

The selection of the best-performing precipitation dataset was evaluated over a reference period using the single metric of the KGE criterion. This criterion is considered to be a good metric as it weighs bias, correlation and RMSE between simulation and observations, all rightfully considered to be important attributes of a good hydrological simulation. There are, however, many other metrics that could have been chosen to perform this comparison, some of which might be even more important for specific applications such as floods. For example, the JRA55 and CFSR reanalyses were at the bottom of the list of the best-performing datasets presented here. However, in other studies, JRA55 was shown to provide the best reanalysis (Odon et al., 2019), while CFSR was successfully used for precipitation modeling (Khedhaouiria et al., 2018). Clearly, the results presented in this paper should only be used as intended (i.e., to study uncertainty related to the choice of a reference climate

dataset), and not as a judgment of the absolute performance of each dataset. As mentioned earlier, it is important to keep in mind that all of the datasets used in this paper generate adequate streamflow simulations.

It is recommended that reference dataset uncertainty be included in climate change impact studies, and especially so in regions with a sparse network of weather stations. We believe that climate dataset uncertainty can be minimized for most
streamflow metrics using a careful validation and selection of the best-performing ones. A dataset ensemble should nonetheless be retained to assess the sensitivity of the impact study to the choice of a reference dataset. As is the case for most other elements of the hydroclimatic modeling chain of future climate change impacts, there is 'no free lunch' in the sense that there is no single recipe, which will be applicable in all cases. Climate dataset performance is spatially-dependent, as shown here and other studies, and will depend on the criteria used to assess said performance. In addition, the relative uncertainty contribution also
depends on the catchment location and streamflow metric under study. The importance of first-order interactions in variance analysis, and especially of interactions between precipitation datasets with GCMs and with the hydrology models testify to the complex nature of the propagation of uncertainties in the hydroclimatic modeling chain. The use of an appropriate credibility climate dataset ensemble is therefore more than likely to be catchment-related and metric-dependent, and some minimum level of upstream validation would be needed to select the best components.
Some level of guidance for impact modelers can nonetheless be offered from the results of this work. Precipitation is the key driver of dataset uncertainty and should therefore be evaluated in climate change studies alongside the more traditional sources of uncertainty. In cases where it is not possible to select multiple precipitation datasets, the results presented in Figure 3 and in Tarek et al. (2020a) indicate that MSWEP merged product dataset is the best performing one, with CHIRPS and ERA5 being the next best. The gauged-only based products were clearly not the best-performing ones over Africa in contrast to a similar
study performed over North-America (Tarek et al., 2020b). This performance ranking is however only based on the KGE calibration metric. While the KGE is a good overall performance metric, it is possible that using a different performance metric might affect this ranking. Streamflow data also come with many potential quality issues that must be taken into consideration (e.g. Tomkins, 2014; Hamilton and Moore, 2012). However, in the overwhelming majority of cases, there are no competing streamflow datasets from which to study uncertainty from. But flawed streamflow records will impact hydrological model
calibration and performance, and may therefore indirectly contribute to hydrological model uncertainty.

## 6 Conclusions

The main objective of this study was to assess the uncertainty related to the choice of a reference dataset against that of other sources of uncertainty in climate change impact studies. This was achieved by performing a large-sample hydrological climate change impact study over 1145 African catchments. The study used 9 precipitation and 2 temperature datasets, along with
10 GCMs and 2 hydrological models, for a total of 360 possible combinations. Temperature dataset-related uncertainty was minimal; with a median relative contribution to uncertainty, less than 0.25% for all 6 presented streamflow metrics. On the other hand, the nine precipitation dataset ensembles generated a future uncertainty equal to or larger than that related to GCMs. Using a reduced ensemble of the best-performing precipitation datasets systematically reduced the precipitation dataset uncertainty,

but still accounted for 10 to 20% of the total variance for 5 of the 6 streamflow metrics, and still remained the main source

of uncertainty for the low-flow metric. The main conclusion of this study is that the choice of a climate reference dataset can induce significant uncertainty in climate change impact studies, at least in regions with a sparse weather station coverage.

*Code and data availability.* The NAC$^2$H climate and streamflow data was downloaded from the NAC$^2$H database, available here: https://osf.io/s97cd/ (Arsenault et al., 2020).

The GRDC streamflow data can be downloaded from the Global Runoff Data Centre, available here: https://portal.grdc.bafg.de/applications/

public.html?publicuser=PublicUser#dataDownload/Home.

ERA-Interim data are available through the ECMWF servers at: https://apps.ecmwf.int/datasets/data/interim-full-daily/ (Dee et al., 2011).

ERA5 data is available on the Copernicus Climate Change Service (C3S) Climate Data Store: https://cds.climate.copernicus.eu/#!/search?text=ERA5&type=dataset (Hersbach and Dee, 2016).

JRA55 datasets are available on the Research Data Archive: https://rda.ucar.edu/datasets/ds628.0/ (Kobayashi et al., 2015).

NCEP datasets can be downloaded from The NCAR Research Data Archive, available here: https://rda.ucar.edu/datasets/ds093.1/#!access (Saha et al., 2010).

MSWEP data are available through the PCA servers at: https://platform.princetonclimate.com/PCA_Platform/mswepRetro.html (Beck et al., 2017a).

The CHIRPS satellite dataset can be downloaded from the Climate Hazards Center: https://www.chc.ucsb.edu/data/chirps (Funk et al., 2015).

The CPC and GPCC datasets can be downloaded from The NOAA Physical Sciences Laboratory (PSL), available here: https://psl.noaa.gov/data/gridded/tables/precipitation.html (Chen et al., 2008; Schneider et al., 2018).

The Coupled Model Intercomparison Project Phase 5 (CMIP5) data can be downloaded from the World Climate Research Programme (WCRP) at: https://esgf-node.llnl.gov/search/cmip5/ (Taylor et al., 2012).

The HMETS hydrological model is available on the Matlab File Exchange:

https://www.mathworks.com/matlabcentral/fileexchange/48069-hmets-hydrological-model (Martel et al., 2017).

Finally, the GR4J model and Cemaneige snow module are made available by the IRSTEA: https://webgr.irstea.fr/en/models/ (Perrin et al., 2003).

*Author contributions.* MT performed all of the computing work, including all datasets and GCMs download, hydrological modelling and

calibration. He performed most of the analysis and wrote the main sections of the paper. FB contributed to experiment design and data analysis and provided multiple edits to the document. RA, provided expertise on the catchment database and parallel computing.

*Competing interests.* The authors declare that they have no conflict of interest.

*Financial support.* This study has been partly funded by the Egyptian Armed Forces (Ministry of Defense). The Natural Sciences and Engineering Research Council of Canada (NSERC) also partly funded this project through François P. Brissette and Richard Arsenault's respective discovery grants (grant nos. RGPIN-2015-05048 and RGPIN-2018-04872).

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

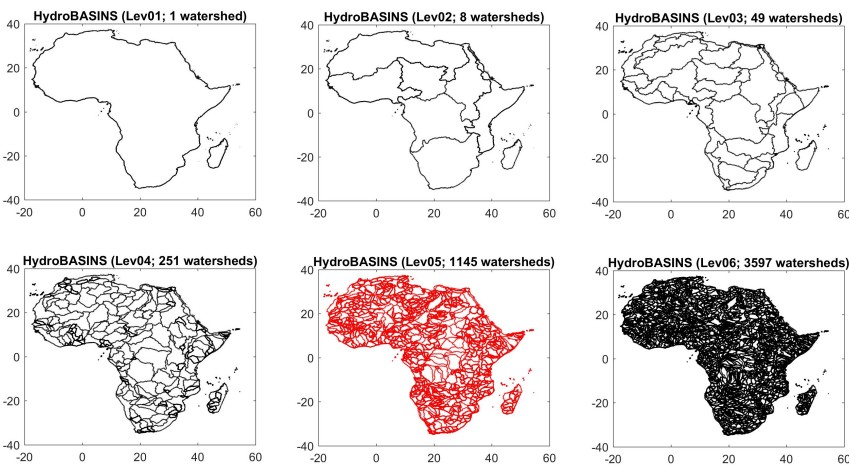

**Figure 1.** Sample of the different vector layers of watersheds on the African continent. Each layer has a different number of watersheds, depending on the required scale.

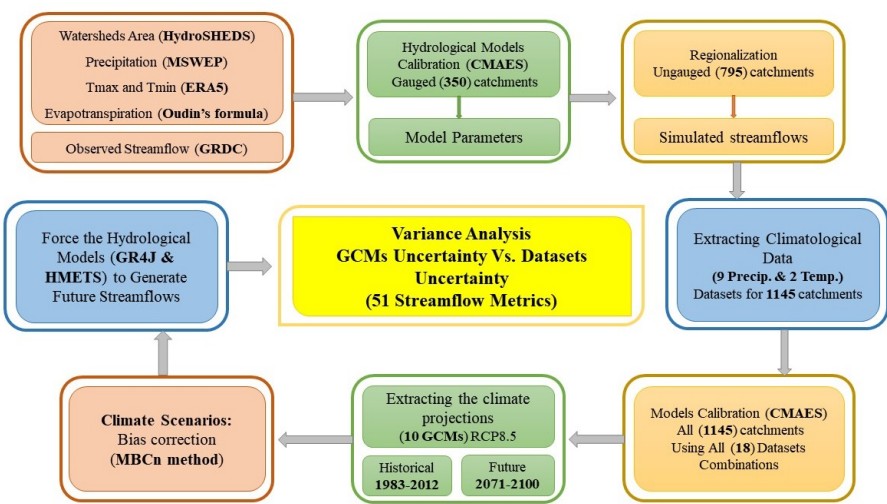

**Figure 2.** Overview of the various methodological steps implemented in this study.

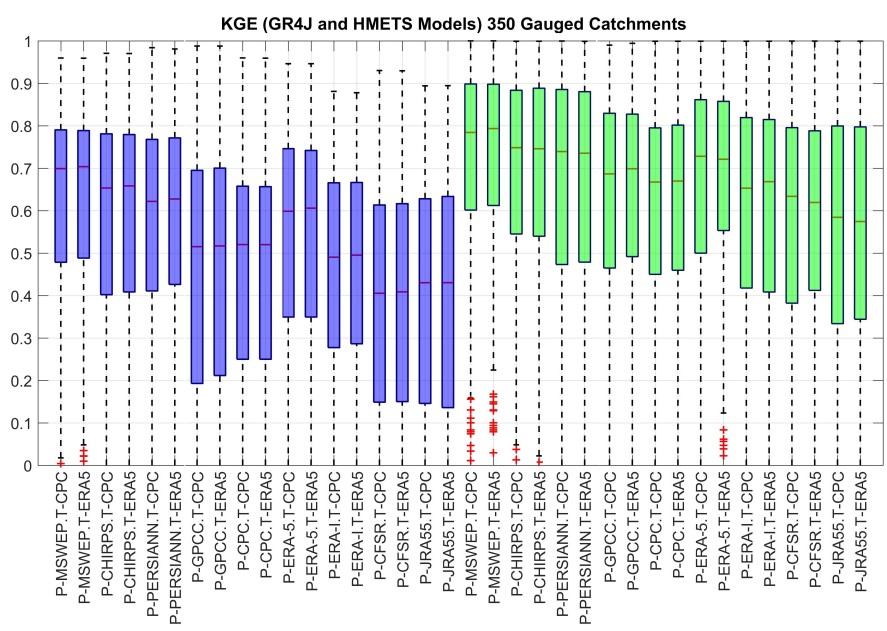

**Figure 3.** KGE calibration values using the 18 possible combinations of precipitation and temperature datasets, for both hydrological models (GR4J in blue and HMETS in green) for each of the 350 selected gauged catchments.

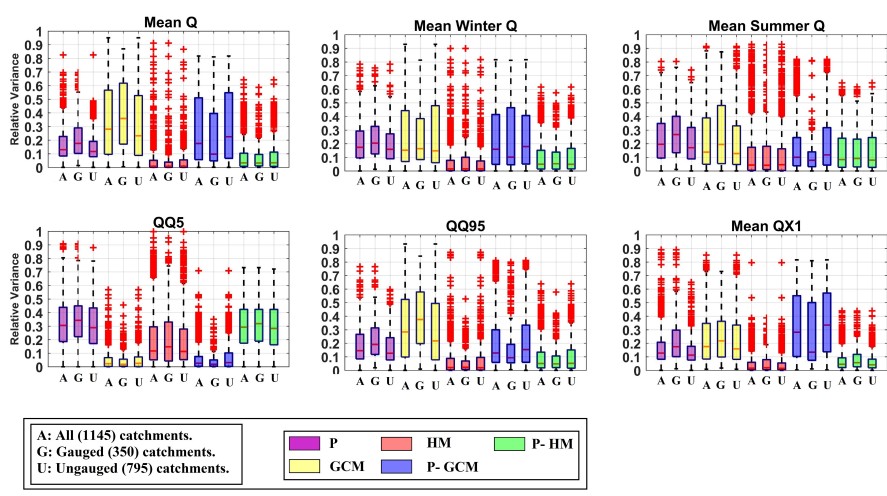

**Figure 4.** Boxplots of the relative variance attribution results for the five main contributors to overall variance (P, GCM, HM, P-GCM and P-HM) and 6 streamflow metrics. Relative variance is shown for all 1145 catchments: (A), 350 gauged (G) and 795 ungauged (U) catchments.

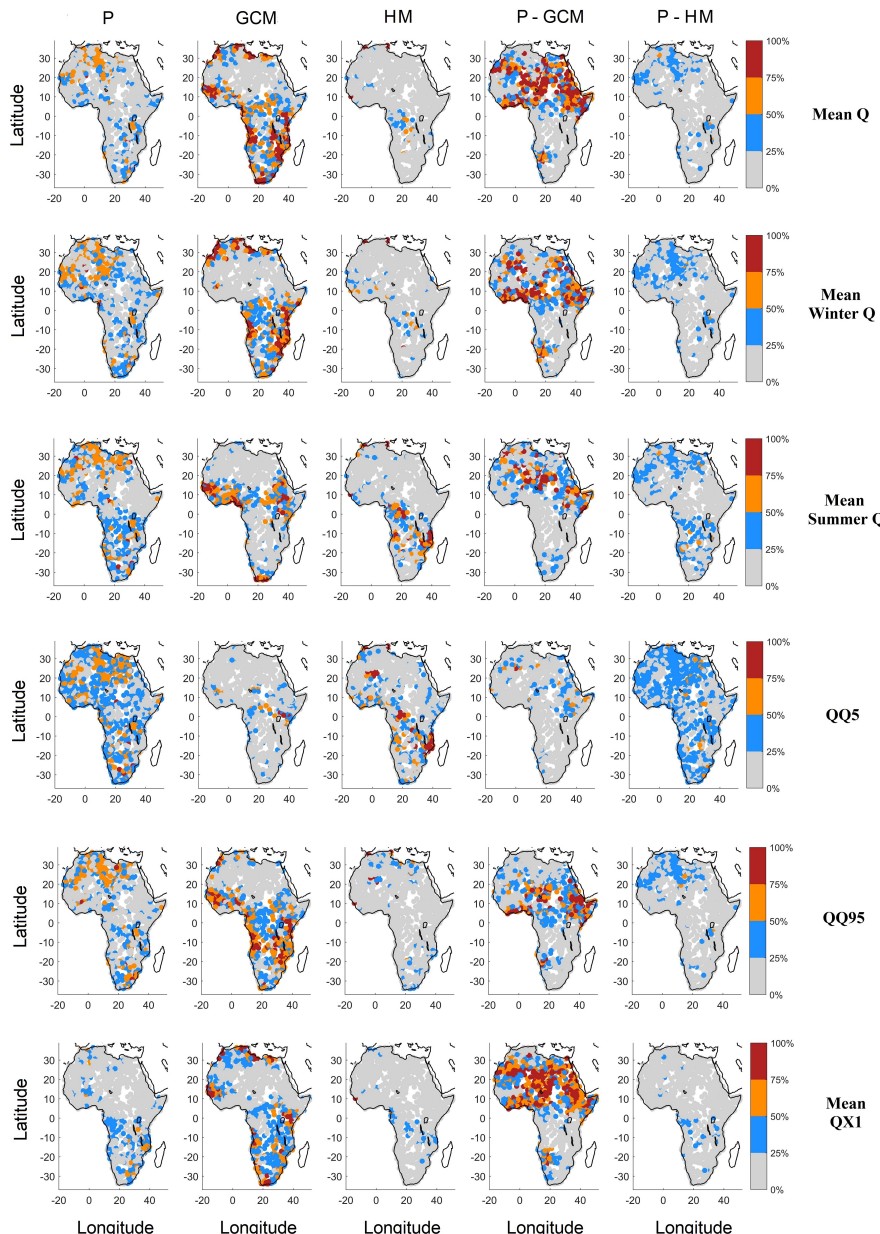

**Figure 5.** Spatial distribution of the five main contributors to variance (columns; precipitation (P), GCMs (GCM), hydrological models (HM), interactions between precipitation datasets and GCMs (P-GCM), and interactions between precipitation datasets and hydrological models (P-HM)) for each of the 6 streamflow metrics (rows; Mean Q, Winter Q, Summer Q, the $5^{th}$ and $95^{th}$ quantiles of streamflow distribution (QQ5 and QQ95, respectively) and the 30-year mean of the annual daily maximum streamflow (QX1)). Each dot represents the watershed centroid.

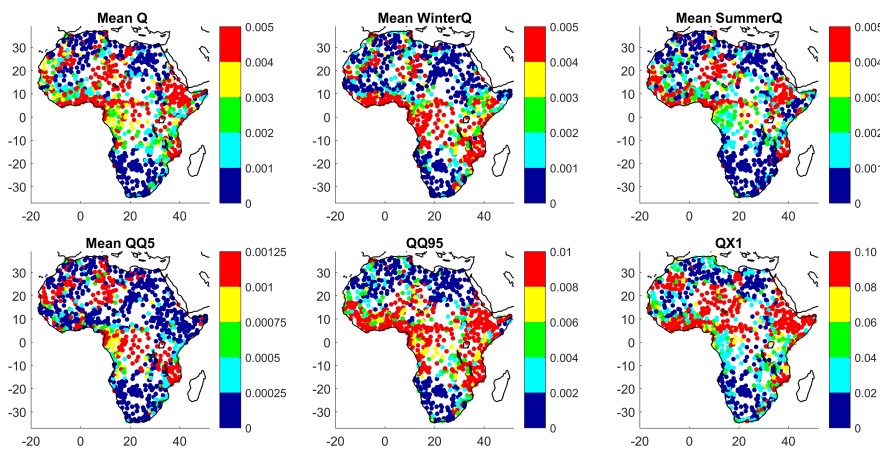

**Figure 6.** Standard deviation of discharge per unit area (in $m^3/sec/km^2$), constructed from 360 values for each catchment and streamflow metric.

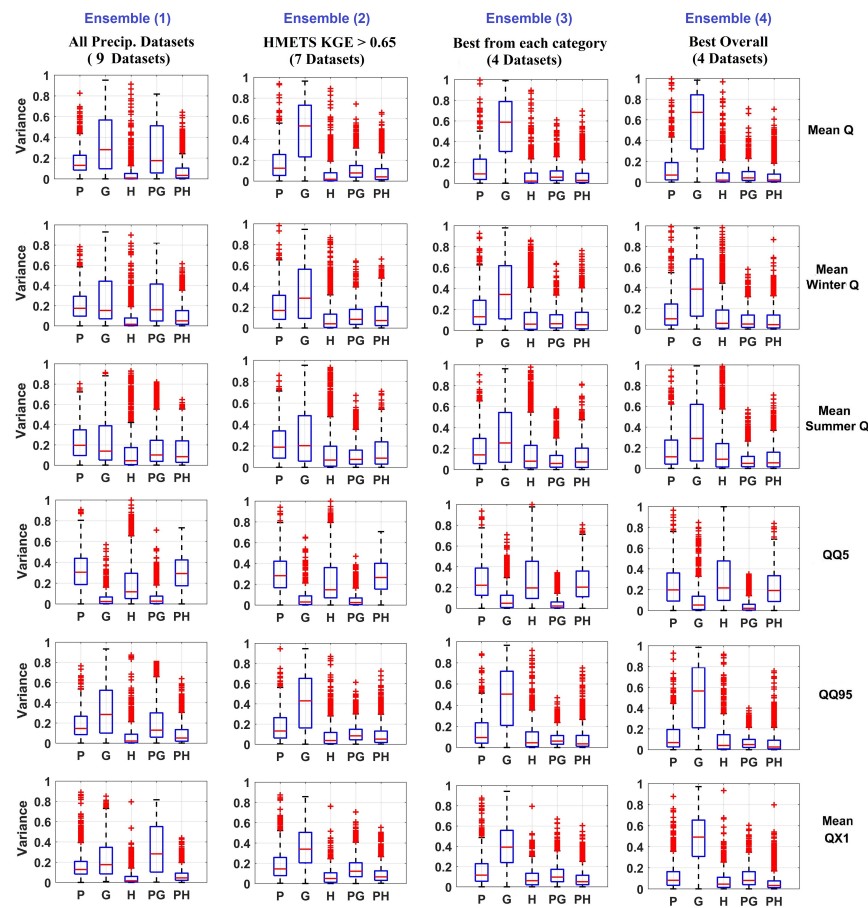

**Figure 7.** Boxplots of the five main components of the variance attribution: precipitation (P), GCMs (G), hydrological models (H), interaction between precipitation datasets and GCMs (PG) and interaction between precipitation datasets and hydrological models (PH). Columns represent the four precipitation ensembles of Table 4, while rows represent the 6 hydrological indices investigated in this study.

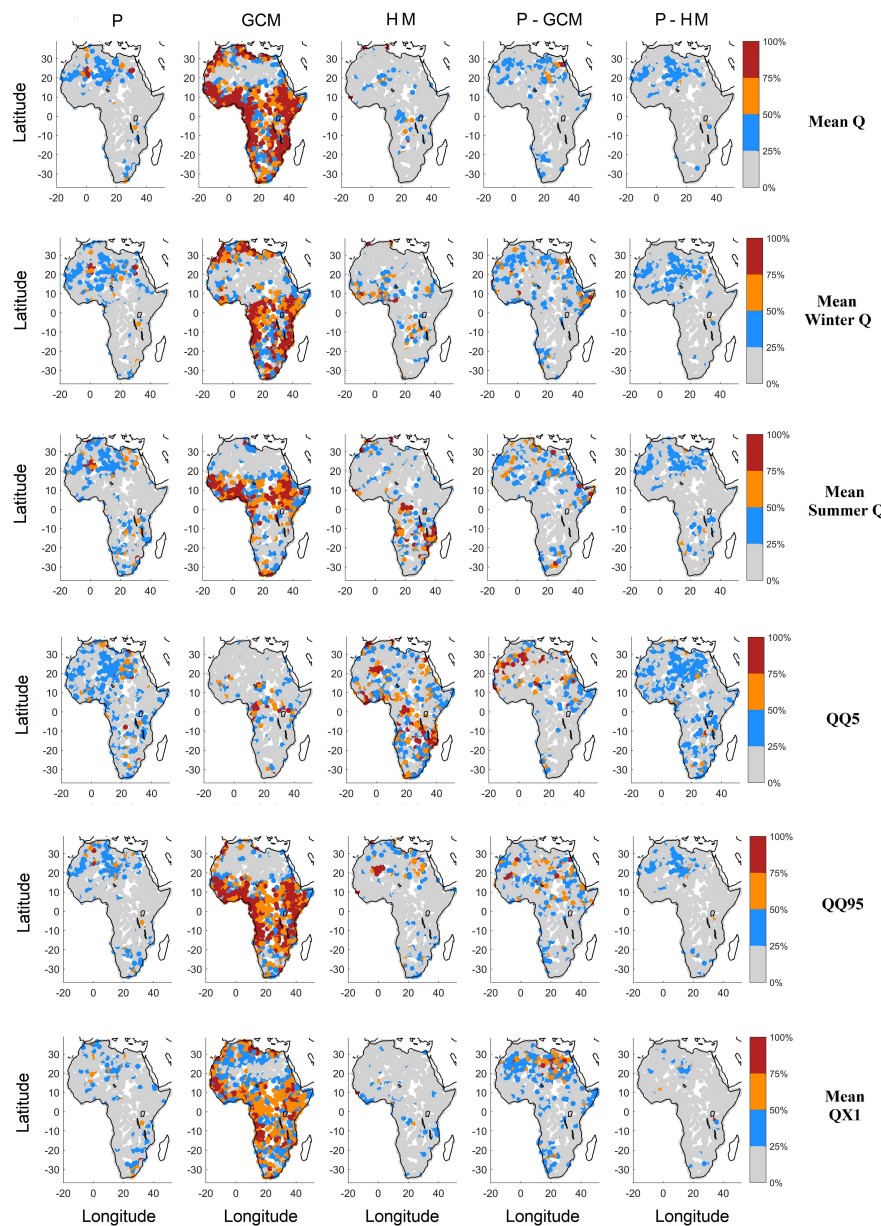

**Figure 8.** Spatial distribution of the five main contributors to variance (columns) for each of the 6 streamflow metrics (rows), using the 4 best precipitation datasets (Ensemble 4 of Table 4). Each dot represents the watershed centroid.

**Table 1.** List of chosen GCMs, research centres and spatial resolutions.

| No. | Models | Research Center | Spatial Resolution |
|-----|--------|-----------------|--------------------|
| 1 | BCC-CSM1-1 | Beijing Climate Center, China Meteorological Administration, China | 2.79° x 2.81° |
| 2 | BNU-ESM | College of Global Change and Earth System Science, Beijing Normal University, China | 2.79° x 2.81° |
| 3 | CanESM2 | Canadian Center for Climate Modeling and Analysis, Canada | 2.79° x 2.81° |
| 4 | CCSM4 | National Center of Atmospheric Research, USA | 0.94° x 1.25° |
| 5 | CMCC-CESM | Centro Euro-Mediterraneo per I Cambiamenti Climatici, Italy | 3.44° x 3.75° |
| 6 | CNRM-CM5 | National Center of Meteorological Research, France | 1.40° x 1.40° |
| 7 | FGOALS-g2 | LASG, Institute of Atmospheric Physics, Chinese Academy of Sciences, China | 2.79° x 2.81° |
| 8 | INMCM4 | Institute for Numerical Mathematics, Russia | 1.5° x 2.0° |
| 9 | MIROC5 | Atmosphere and Ocean Research Institute (The University of Tokyo), National Institute for Environmental Studies, and Japan Agency for Marine-Earth Science and Technology, Japan | 1.40° x 1.40° |
| 10 | MRI-CGCM3 | Meteorological Research Institute, Japan | 1.12° x 1.125° |

**Table 2.** The selected global gridded datasets.

| No. | Short Name | Data Source | Spatial Resolution | Spatial Coverage | Temporal Resolution | Temporal Coverage |
|---|---|---|---|---|---|---|
| | | | 1- Precipitation datasets | | | |
| 1 | CPC Unified | Gauge | 0.5° | Global | Daily | 1979- Present |
| 2 | GPCC | Gauge | 1.0° | Global | Daily | 1982- 2016 |
| 3 | PERSIANN-CDR (V1R1) | Gauge, Satellite | 0.25° | ±60° Lat. | 6 hourly | 1983- 2012 |
| 4 | CHIRPS V2.0 | Gauge, Satellite | 0.05° | ±50° Lat. | Daily | 1981- Present |
| 5 | NCEP-CFSR | Reanalysis | 0.5° | Global | 6 hourly | 1979- 2012 |
| 6 | ERA-Interim | Reanalysis | 0.75° | Global | 3 hourly | 1979- 8/2019 |
| 7 | ERA5 | Reanalysis | 0.25° | Global | hourly | 1979- 2017 |
| 8 | JRA-55 | Reanalysis | 0.5625° | Global | 3 hourly | 1959- Present |
| 9 | MSWEP V1.2 | Gauge, Satellite, Reanalysis | 0.25° | Global | 3 hourly | 1979- 2015 |
| | | | 2- Temperature datasets | | | |
| 1 | CPC Unified | Gauge | 0.5° | Global | Daily | 1979- Present |
| 2 | ERA5 | Reanalysis | 0.25° | Global | hourly | 1979- 2017 |

**Table 3.** Mean percentage of variance for 6 streamflow metrics for 1145 catchments. All main effects (P, GCM, temperature (T), and HM) and first-order interactions are shown in rows 3 to 12. The last row sums up the second- and third-order elements contribution to variance. QQ5 and QQ95 are respectively the $5^{th}$ and $95^{th}$ quantiles of streamflow distribution. QX1 is the 30-year mean of the annual daily maximum streamflow value. These rows written in **bold font** outline the main contributors to variance.

| | Mean relative variance (%) | | | | | | |
|---|---|---|---|---|---|---|---|
| | Mean Q | Winter Q | Summer Q | QQ5 | QQ95 | QX1 | Average |
| **P** | **21.62** | **24.12** | **28.54** | **34.38** | **23.17** | **22.36** | *25.70* |
| **GCM** | **39.71** | **24.93** | **27.29** | **4.39** | **39.56** | **25.82** | *26.95* |
| T | 0.17 | 0.12 | 0.09 | 0.02 | 0.15 | 0.04 | 0.09 |
| **HM** | **5.18** | **8.43** | **19.99** | **21.96** | **5.59** | **5.50** | *10.11* |
| **P-GCM** | **21.55** | **25.19** | **10.20** | **3.42** | **16.01** | **26.33** | *17.12* |
| P-T | 0.02 | 0.01 | 0.02 | 0.01 | 0.02 | 0.01 | 0.015 |
| **P-HM** | **7.38** | **9.72** | **14.69** | **31.12** | **8.17** | **8.78** | *12.31* |
| GCM-T | 0.01 | 0.01 | 0.006 | 0.0018 | 0.017 | 0.005 | 0.008 |
| GCM-HM | 1.30 | 2.13 | 1.44 | 1.36 | 2.49 | 3.49 | 2.04 |
| T-HM | 0.0087 | 0.0098 | 0.0069 | 0.0041 | 0.0189 | 0.0058 | 0.009 |
| Others | 2.78 | 5.20 | 3.46 | 2.99 | 4.60 | 7.58 | 4.43 |

**Table 4.** List of ensemble of precipitation datasets.

| Ensemble no. | Number of precipitation datasets | Rationale for selection | Datasets included | Datasets excluded |
|---|---|---|---|---|
| 1 | 9 | All 9 | All | None |
| 2 | 7 | Mean KGE $\geq$ 0.65 | MSWEP, GPCC, CPC, CHIRPS, PERSIANN ERA5 and ERA-I | CFSR and JRA55 |
| 3 | 4 | Best of each category (merged, satellite, gauge, and reanalysis) | MSWEP, CHIRPS, GPCC and ERA5 | CPC, PERSIANN, ERA-I, CFSR and JRA55 |
| 4 | 4 | Best (4) | MSWEP, CHIRPS, PERSIANN and ERA5 | CPC, GPCC, ERA-I, CFSR and JRA55 |