# Peer review of "Uncertainty of gridded precipitation and temperature reference datasets in climate change impact studies"

_Hydrology and Earth System Sciences, 2020_

## Short Comment (SC1) · 28 Oct 2020

I found this study really informative about the issues related to the choice of "observed" rainfall for hydrological impact studies of climate change.

However, the regionalization approach considered here is barely described in section 3.2, when the results for 795 stations out of 1145 rely on this regionalization procedure. I am not aware of other studies attempting to regionalize the model parameters of the GR4J or HMETS models at the scale of Africa, so the results of this regionalization procedure surely deserve more than one sentence (line 202). In fact, I really believe this type of regional analysis would requires a study on its own. What is the efficiency of

the regionalization methods tested in a cross-validation framework? Beside the spatial proximity, how are the "physical similarity" and the "multiple-linear regression" methods implemented? what are the predictors, since the authors only mention watershed delineation in the manuscript?

Another aspect is the presence of dams and reservoirs. Many African rivers are regulated and no mention is given in the data section 2.2.3 if the selected rivers are regulated or not. We recently released a large dataset or river discharge in Africa (https://doi.org/10.23708/LXGXQ9) and from the metadata it can be seen that about one third of the basins are regulated. It could explain the bad modelling results for some basins, since the hydrological models are not validated against independent data in the present study (line 182). It is likely that the modelling results with different satellite products can be impacted by river regulation in some basins.

Finally, I am a bit surprised by the figures 5, 6 and 8, you have river runoff over the Sahara Desert?

---

## Referee Comment (RC1) · Anonymous Referee #1 · 9 Dec 2020

In sparsely gauged regions, satellite, reanalysis and merged products datasets are valuable for hydro-meteorological studies. However, it is not yet well understood how uncertainty cascades from the choice of a reference dataset down to future climate streamflows in hydroclimatic impact studies. To bridge this gap, the authors assess relative uncertainty contributions from 4 main contributors (element of the hydroclimatic modelling chain; GCMs, precipitation and temperature datasets and hydrological models) for future climate streamflow predictions in across 1145 African catchments. They show that GCMs and precipitation datasets are the main contributors of uncertainty, hinting at the importance of the choice of the modelling chain and reference datasets for hydroclimate impact studies.

[Figure]

This paper is overall well-written and aims to bridge an important gap in hydroclimate impact studies. The methodology used however currently suffers from a few major limitations which should be addressed by the authors (see the first three key comments below). Please find below a few comments which will hopefully help improve the paper for publication.

Key comments

- I wonder about the conservation of the important energy and water balance when combining P and T datasets from different sources - e.g. MSWEP precipitation and ERA5 temperature. Could you please comment on this and potential impacts in the paper?

- On P10 L280-281 you mention the impacts of the reduction in the ensemble size of the precipitation dataset on the variance analysis: "Unsurprisingly, it shows that reducing the size of the precipitation ensemble results in a consistent decrease in the variance attributed to precipitation". This leads me to question what the impacts may be of the ensemble sizes from the contributors you investigate on your conclusions: 10 GCMs, 2 hydrological models, 2 temperature and 9 precipitation datasets. Wouldn't it be more adequate to have the same number of ensemble members coming from these various components of the chain? For example, temperature appears to play a minor role in the analysis, however, only 2 members were used here, which could impact this conclusion somewhat. Please reflect on this in your paper.

- The calibration strategy may be problematic for this study given the climate timescales explored. As mentioned by Arsenault et al. (2018): "In this study, the effect of calibration and validation is investigated on three catchments that did not show signs of non-stationarity, i.e. the mean annual streamflow did not contain a trend over a 25-year period. This allowed randomly sampling from the database to generate calibration and validation sets. This raises the question as to how the method would fare on a catchment that is subject to non-stationarity. Obviously, in this scenario, the independent

test period would need to be in the most recent years and those years could not be randomly selected from the entire time series." Arsenault et al. go on to suggest alternative methods which could be used in such catchments. Could you please reflect on the adequacy of this calibration strategy for your study?

- You mention 51 different streamflow metrics, yet all results are shown for only 6 metrics. Could you please comment on the results from the additional metrics not shows here? These could perhaps go in as supplementary material?

- Your figures are very rich in results. Please guide the readers a bit more by mentioning what is shown in the columns/rows, etc. when introducing each figure (especially for Fig 5, 7 and 8).

Minor comments

- P1 L9-12: Please clarify here if these datasets are deterministic or ensembles.

- P1 L15: Please explain here what CMIP5 GCMs stands for.

- P3 L60: Please clarify here what GHGES stands for, it only comes later on L66.

- P3 L63: Could you please give readers a brief explanation of what the "change factor approach for downscaling" is here?

- P4 L91: Since you are looking at hydroclimate impacts, it might make sense to also refer to the Hydrological Climate Classification by Knoben et al. (2018), more adapted to hydrological studies: Knoben, W. J., Woods, R. A., & Freer, J. E. (2018). A quantitative hydrological climate classification evaluated with independent streamflow data. Water Resources Research, 54(7), 5088-5109, https://doi.org/10.1029/2018WR022913.

- P4 L107: Please clarify here what NAC2H stands for.

- P4 L112: Please consider replacing "(or better)" with "or higher".

- P5 L122: While it is implicit, you do not actually explicitly mention that you have used

GRDC data in this paper.

- P5 L136: Could you please provide your reasoning for selecting the L5 vector layer instead of the other 3 shown on Fig. 1?

- P5 L137: I would find it helpful if you could briefly go over the steps of the entire hydroclimatic modelling chain from Fig. 2 in the text as well. For example, I only found out from Fig. 2 that two calibrations are performed.

- P5 L146: Please summarise what the "4 groups of components of the uncertainty modeling chain" are here for added clarity for the readers.

- P6 L150-152: Could you please summarise briefly in the text as well which metric(s) and time period were used for the calibration? I read the added information later on in Section 3.1.3, please mention here that more details are given in that later section.

- P6 L155-158: This is a repetition to an extent of P6 L147-149. Please consider merging these two paragraphs for more clarity. It is also unclear to me how the 1150 African stations (L155) became 1145 catchments?

- P6 L162-163: More importantly, have they been shown to perform well in Africa specifically?

- P7 L201-204: Is this single simulation used as a reference against which to verify the other simulations produced as part of the analysis? Please clarify as it confused me a little bit. When you say "Based on the hydrological modeling performance on the 350 gauged catchments", do you mean the calibration performance? Please clarify here.

- P8 L213: I would have liked to read a bit more about the variance analysis, about the methods and the aim of this analysis. E.g. What are the variance components and what do they tell you? Is such an analysis computationally expensive to run?

- P8 L214-216: Could you please provide an overview of the metrics computed, perhaps in a table?

- P8 L232-233: Please clarify that this observation is with regards to calibration for these 350 catchments. It could otherwise be misinterpreted taken out of context.

- P9 L256-256: The mean Summer Q also appears to show a different signal from the other metrics.

- P9 L260-261: This is arguable and quite complicated to see. Perhaps putting the metrics in columns and the contributors in rows might help see these better?

- P9 L268-269: Which variance contributor is Fig. 6 shown for?

- P10 L284-285: It seems to be that most of the drop is seen between ensembles 1 and 2, rather than 2 and 3. Please also add the ensemble numbers 1-4 from table 4 to Fig. 7.

- P10 L290-292: Hydrological model uncertainty appears dominant over precipitation uncertainty for low flows.

- Figure 2: In order to be clearer for the readers, please consider adding clear sub-heading for each box in this diagram.

---

## Referee Comment (RC2) · Anonymous Referee #2 · 23 Dec 2020

The manuscript by Tarek et al compares nine precipitation and two temperature datasets for 1145 African catchments to understand the uncertainties they introduce when used in hydrological modelling applications. The study is substantial, addresses a significant community challenge, and offers practical guidance to others in the field. On that basis I judge that it will be of clear interest to the readership of HESS, with substantial impact in Africa and insights applicable more widely. There are three major points which the authors should address in revision, and some minor points which they might wish to consider too:

Major points

1. Bias correction is applied but only five lines are given to the description of the method used. It is essential to have more information on the bias corrections made, especially at rarely seen precipitation quantiles, given the sparsity of data in this region.

2. Hydrological modelling uncertainty (L240ff): more information is needed to attribute uncertainty in the hydrological modelling part of the work. It is unclear at present which components of the hydrological models contribute greatest uncertainty, especially with the HMETS model.

3. Applicabilty (L360 onwards): one of the key virtues claimed in the introduction is the guidance offered to future researchers tackling this problem afresh. Yet in this section the reader is left with the sense that there is no clear advantage to any particular dataset, with conclusions left as context dependent. Clearly that's a valid finding but it needs to be reflected in the stance that the paper offers in its opening and concluding paragraphs.

Minor comments / editorial points

Reference formatting is inconsistent with many additional brackets, etc. L173: hypodermic -> subsurface Figure 5: annotations and labels are hard to read; please enlarge

---

## Author Response (AR1)

**Response to Reviewers' Comments**

We appreciate the efforts of the reviewers and we thank them for their insightful and constructive comments. We have provided information points and clarifications as to how we will modify the manuscript accordingly below. We provide detailed responses to each of the reviewers' comments. For convenience, we put the reviewer comments in black font, and author responses in blue.

**RC1- Anonymous Referee #1 comments:**

This paper is overall well-written and aims to bridge an important gap in hydroclimate impact studies. The methodology used however currently suffers from a few major limitations which should be addressed by the authors.

We wish to thank you for your time reviewing our paper. We are confident that a modified version of this paper will address all of your comments in a satisfactory manner.

**Key comments**

-  I wonder about the conservation of the important energy and water balance when combining P and T datasets from different sources - e.g. MSWEP precipitation and ERA5 temperature. Could you please comment on this and potential impacts in the paper?

This is a very interesting comment, but a complex issue with many different angles. In most cases when doing hydrological modeling, energy and water balance is not taken into account by the driving datasets, so we don't believe that this is a problem specific to our work. Most data used in hydrological modeling comes from gridded information and the gridding process is almost entirely done independently for precipitation and temperature, therefore not even taking into account basic temporal correlations between both variables.  Even in hydrological models that use station data, there is usually some level of interpolation/extrapolation to extend the information at the catchment scale. Most precipitation products are now developed independently of temperature, as is the case for most dataset used in this study. Reanalyses are the most consistent dataset with respect to energy budget and water balance. However, even though the weather model of the reanalysis is entirely physically coherent, the data assimilation does not preserve this physical coherency and therefore reanalysis does not fully conserve water balance. And as shown in this study (and others), reanalysis precipitation, although much improved, is not as good as other precipitation datasets. We will definitely discuss these issues in a revised discussion since they are definitely relevant.

*"Combining different - and somewhat independent - data sources for temperature and precipitation raises potential issues about mass and energy balance. Most of the products used*

*in this work originate from a gridding process that is independently done for precipitation and temperature, therefore not taking into account temporal correlations between both variables. Most precipitation products are also developed independently of temperature. Reanalyses are the most consistent dataset with respect to energy budget and water balance. However, even though the weather model of the reanalysis is entirely physically coherent, the data assimilation does not preserve this physical coherency and therefore reanalysis does not conserve water balance. The combination of precipitation and temperature datasets is therefore unlikely to be problematic."*

- On P10 L280-281 you mention the impacts of the reduction in the ensemble size of the precipitation dataset on the variance analysis: "Unsurprisingly, it shows that reducing the size of the precipitation ensemble results in a consistent decrease in the variance attributed to precipitation". This leads me to question what the impacts may be of the ensemble sizes from the contributors you investigate on your conclusions: 10 GCMs, 2 hydrological models, 2 temperature and 9 precipitation datasets. Wouldn't it be more adequate to have the same number of ensemble members coming from these various components of the chain? For example, temperature appears to play a minor role in the analysis, however, only 2 members were used here, which could impact this conclusion somewhat. Please reflect on this in your paper.

This is a fair question, and it goes to the heart of the uncertainty issue. In an ideal world, it would be best to have a similar number of ensemble members for each uncertainty component. However, the contribution to variance is related to how dissimilar the ensemble members are and not strictly to its numbers. As such, the ensemble with the fewest members can still provide the largest contribution to variance. If you start with a single member (a single hydrological model for example), each additional hydrological model added to the ensemble will add uncertainty. However, there is a point of diminishing return where adding more hydrology models will not change anything because the added hydrological response will be within an envelope defined by the existing models. So ultimately, we don't need all ensembles to have the same number of members, we need all ensembles to have enough credible members to cover the uncertainty. For example, based on the two temperature datasets used here, adding more temperature datasets is unlikely to change the results considering how little uncertainty is present in the two datasets when compared to other sources. Adding more hydrological models is likely to have a much more important impact. Based on previous published work, 10 GCMs is more than enough to frame the uncertainty contribution from this source. I think expanding this discussion in the revised version of the manuscript would be useful.

*"The number of components in a variance attribution study is an important issue. However, the contribution to variance is related to how dissimilar the ensemble members are and not strictly to its numbers. As such, the ensemble with the fewest members can still provide the largest contribution to variance. There is therefore no need for all ensembles to have the same number of members, but rather to have enough credible members to cover the uncertainty. Despite having only two temperature datasets here, adding more temperature datasets is unlikely to change the results considering how little uncertainty is present in the two datasets when compared to other sources. Temperature is the easiest variable to measure and to extrapolate, especially when compared to precipitation. It is therefore expected that precipitation uncertainty would normally*

*dwarf the contribution of temperature. Based on previously published work, 10 GCMs is very likely more than enough to frame the uncertainty contribution from this source (e.g. Wang et al., 2020)."*

\-        The calibration strategy may be problematic for this study given the climate timescales explored. As mentioned by Arsenault et al. (2018): "In this study, the effect of calibration and validation is investigated on three catchments that did not show signs of non-stationarity, i.e. the mean annual streamflow did not contain a trend over a 25-year period. This allowed random sampling from the database to generate calibration and validation sets. This raises the question as to how the method would fare on a catchment that is subject to non-stationarity. Obviously, in this scenario, the independent test period would need to be in the most recent years and those years could not be randomly selected from the entire time series." Arsenault et al. go on to suggest alternative methods which could be used in such catchments. Could you please reflect on the adequacy of this calibration strategy for your study?

This is a good question, which has been debated previously so we completely understand the idea behind the comment. In the methods, we describe that the catchments were calibrated on the full length of the available data. The first reason is to ensure as much information as possible is contained in the parameter set, and the second is to maximize the chances of regionalization methods to perform well for the ungauged catchments. These two objectives can be seen as mostly in the same direction, but there is a twist that makes them contradictory in one sense, and the approach that we implemented is somewhat of a compromise on these issues.

There are two main concepts here that need to be highlighted:

1-    In Arsenault et al. (2018), the main conclusion is that the more years are used in calibration, the more the parameter set contains useful information as it can compensate and accommodate for more types of events. In other words, if one were to calibrate on a single, dry year, then the validation would probably be atrocious for a very humid year. But by calibrating on a series of wet and dry years, the parameter set can compromise and be good on the entire period. The same concept is applied here, where we want to keep as much information as possible in the parameter set so as to make any relationships between catchment descriptors and parameters (if any) as robust as possible for regionalization. The same also holds for simulation, in that in the absence of any knowledge a priori of the impacts of climate change, using the entire parameter set is prudent as it protects against highly variable changing conditions in the future. It might be possible to identify non-stationarities and target them specifically, but that would then mean that in regionalization, these parameters would also likely perform much worse due to the limited information contained within. Therefore, it was decided to maximize the information content and ensure that the widest possible "spread" of conditions was included in the calibration data. This was done by taking the entire period.

2-    In regionalization, we have the advantage that the "verification" set is actually the pseudo-ungauged catchment itself. Indeed, the "validation" on the donor catchments is not really useful since the score we want to improve is the regionalization skill on an

independent catchment, with it's own data coverage period. So in this case the calibration is done on the donor catchments, and the "validation" is done on the pseudo-ungauged basins. Keeping years as "validation" years on the donor catchments (by split-sample calibration or other) is counter-productive because:

> a. There are fewer years to build the parameter-descriptor relationships;
>
> b. The donor and ungauged catchments have different periods, which means that any difference in period between the gauged and ungauged basins could artificially increase (or decrease) the apparent effect of non-stationarities.

We understand that this was not at all detailed in the previous version of the paper, therefore we will flesh it out more in the next version to ensure that the rationale is well described.

*"The Arsenault et al. (2018) study was performed on catchments which showed no signs of non-stationarity. We applied the same methodology here despite foregoing any testing for homogeneity. For regionalization purposes, the maximum parameter identifiability was deemed preferable and using a longer time period maximized the likelihood of parameter identifiability. The same also holds for simulation, in that in the absence of any knowledge a priori of the impacts of climate change, using the entire parameter set is prudent as it protects against highly variable changing conditions in the future."*

- You mention 51 different streamflow metrics, yet all results are shown for only 6 metrics. Could you please comment on the results from the additional metrics not shown here? These could perhaps go in as supplementary material?

We have modified this in the revised version.

*"Analysis of variance was performed for six streamflow metrics out of the fifty-one metrics defined in Arsenault et al. (2020) for each of the 1145 catchments. These six metrics cover a wide range of streamflow conditions: mean annual (Mean Q), seasonal (Winter Q and Summer Q) values, the 5th and 95th distribution quantiles (QQ5 and QQ95, respectively), as well as annual daily extreme (QX1) metrics."*

- Your figures are very rich in results. Please guide the readers a bit more by mentioning what is shown in the columns/rows, etc. when introducing each figure (especially for Fig 5, 7 and 8).

Thank you for the comment, we will ensure that the figures are explained in more detail in the revised manuscript.

-

**Minor comments**

[Page 1, lines 9-12] Please clarify here if these datasets are deterministic or ensembles.

These datasets are deterministic. This will be specified in the revised version.

*"To tackle this issue, this study compares nine precipitation and two temperature datasets over 1145 African catchments to evaluate the dataset uncertainty contribution to the results of climate change studies. These deterministic datasets all cover a common 30-year period needed to define the reference period climate."*

[Page 1, line 15] Please explain here what CMIP5 GCMs stands for.

The acronyms (CMIP5 – fifth Coupled Model Intercomparison Project, GCM- General Circulation Model) will be defined in the revised version.

*"To assess dataset uncertainty against that of other sources of uncertainty, the climate change impact study used a top-down hydroclimatic modeling chain using 10 CMIP5 (fifth Coupled Model Intercomparison Project) General Circulation Models (GCMs) under RCP8.5 and two lumped hydrological models (HMETS and GR4J) to generate future streamflows over the 2071-2100 period."*

[Page 3, line 60] Please clarify here what GHGES stands for, it only comes later on L66.

Thanks for picking this up. We'll make sure the acronym is defined once it's first introduced in the revised version.

*"Rowell (2006) compared the effect of different sources of uncertainty using the initial condition ensembles of different GCMs, Greenhouse Gases Emission Scenarios (GHGES) and Regional Circulation Models (RCMs) on changes in seasonal precipitation and temperature in the United Kingdom."*

[Page 3, line 63] Could you please give readers a brief explanation of what the "change factor approach for downscaling" is here?

We propose removing the reference to the change factor method and simply mention 'using a single downscaling method' as we don't believe a description of the change factor method would be very useful at this stage of the paper.

*"Minville et al. (2008) used ten equally-weighted climate projections derived from a combination of five GCMs, two GHGES and a single downscaling method for downscaling to investigate the uncertainty envelope of future hydrologic variables.*

[Page 4, line 91] Since you are looking at hydroclimate impacts, it might make sense to also refer to the Hydrological Climate Classification by Knoben et al. (2018), more adapted to hydrological studies: Knoben, W. J., Woods, R. A., & Freer, J. E. (2018). A quantitative hydrological climate classification evaluated with independent streamflow data. Water Resources Research, 54(7), 5088-5109, https://doi.org/10.1029/2018WR022913.

That's a good suggestion that will be implemented in the revised version.

*"Looking at the more recent hydrological climate classification of Knoben et al. (2018), Africa can be classified as a no-snow continent, with a strong precipitation seasonality between the tropics and a high aridity index in the extratropical zones, as well as along the coast of the Indian Ocean in the tropical band."*

[Page 4, line 107] Please clarify here what NAC2H stands for.

Good point. This should have been done indeed. NAC2H: The North American Climate Change and Hydroclimatology Data Set.

*"Ten CMIP5 GCMs from 10 different modeling centers were selected for this study, as shown in Table 1. They were selected as a subset of the GCMs used to set up the North American Climate Change and Hydroclimatology (NAC2H) database (Arsenault et al., 2020)."*

 [Page 4, line 112] Please consider replacing "(or better)" with "or higher".

Thanks, we will implement this correction.

*" The precipitation and temperature dataset selection was made on the basis of a high spatial resolution, daily (or higher) temporal resolution, and of the availability of at least 30 years of data covering the same time period, in order to properly define the reference climate."*

[Page 5, line 122] While it is implicit, you do not actually explicitly mention that you have used GRDC data in this paper.

We will correct this in the revised version.

 *"The observed streamflow records were obtained from the Global Runoff Data Centre (GRDC) archive. The GRDC is arguably the most complete global discharge database providing free access to river discharge data (Fekete and Vörösmarty, 2007)."*

[Page 5, line 136] Could you please provide your reasoning for selecting the L5 vector layer instead of the other 3 shown on Fig. 1?

Good point. This was a subject of discussion early on in the course of this work. Ultimately, it was selected as a compromise between having a sensible number of watersheds and keeping the large computational burden of this project reasonable. This will be specified in the revised version.

*"The vector layer (Lev05), which consists of 1145 watersheds, was chosen to be used in this study. It was selected as a compromise between having a sensible number of watersheds and keeping the large computational burden of this project reasonable."*

[Page 5, line 137] I would find it helpful if you could briefly go over the steps of the entire hydroclimatic modelling chain from Fig. 2 in the text as well. For example, I only found out from Fig. 2 that two calibrations are performed.

We will expand the presentation of Figure 2 in the revised manuscript.

[Page 5, line 146] Please summarise what the "4 groups of components of the uncertainty modeling chain" are here for added clarity for the readers.

We will gladly expand the presentation of these components in the revised version.

*"An n-dimensional analysis of variance is performed to partition the uncertainty linked to the four groups of components of the uncertainty modeling chain: precipitation datasets, GCMs, temperature datasets and hydrological models."*

[Page 6, lines 150-152] Could you please summarise briefly in the text as well which metric(s) and time-period were used for the calibration? I read the added information later on in Section 3.1.3, please mention here that more details are given in that later section.

Good point. We will do as suggested.

*"Both hydrological models were calibrated on all catchments for all 18 combinations of reference datasets (2 temperature datasets x 9 precipitation datasets), for a total of 41,220 independent hydrological model calibrations. More details about the calibration process are described later in section 3.1.3."*

[Page 6, lines 155-158] This is a repetition to an extent of P6 L147-149. Please consider merging these two paragraphs for more clarity. It is also unclear to me how the 1150 African stations (L155) became 1145 catchments?

We clearly missed this in the proofreading stage of the manuscript. We will correct this oversight in the revised manuscript.

*"In this study, 350 stations were chosen from the GRDC database based on three criteria. First, stations should have data for the 1983-2012 study period. Second, stations that have less than five consecutive years of data during this period were excluded. Finally, all the stations should be compatible with the selected HydroSHEDS catchments. In order to include additional catchments to allow for a better spatial coverage over the African continent, an additional 795 catchments (the remaining catchments from the Lev05 layer of Figure 1) were selected and an additional regionalization step was performed to generate streamflows at these 795 catchments."*

[Page 6, lines 162-163] More importantly, have they been shown to perform well in Africa specifically?

While these models have been mostly used outside of Africa, there are some cases where the models have been used over Africa (e.g. Essou and Brissette, 2013; Gosset et al.,2013; Simonneaux et al., 2008). Nonetheless, we strongly believe that a successful application in the same climate zone over another continent is certainly a robust enough justification. We will add the above references to the revised version and expand the justification.

Essou, G. R., & Brissette, F. (2013). Climate change impacts on the Oueme river, Benin, West Africa. Journal of Earth Science & Climatic Change, 4(6), 1.

Gosset, Marielle, et al. "Evaluation of several rainfall products used for hydrological applications over West Africa using two high-resolution gauge networks." Quarterly Journal of the Royal Meteorological Society 139.673 (2013): 923-940.

Simonneaux, V., et al. "Modelling runoff in the Rheraya Catchment (High Atlas, Morocco) using the simple daily model GR4J. Trends over the last decades." 13th IWRA World Water Congress, Montpellier, France. 2008.

*"The two hydrological models have been shown to perform well in a wide range of studies and over a wide range of climate zones (Arsenault et al., 2018; Essou & Brissette 2013; Gosset et al., 2013; Martel et al., 2017; Simonneaux et al., 2008; Tarek et al., 2019, 2020b; Valéry et al., 2014)."*

[Page 7, lines 201-204] Is this single simulation used as a reference against which to verify the other simulations produced as part of the analysis? Please clarify as it confused me a little bit. When you say "Based on the hydrological modeling performance on the 350 gauged catchments", do you mean the calibration performance? Please clarify here.

Yes, by hydrological model performance, we mean the calibration performance tested using the KGE objective function. This will be clarified in the revised version.

*"Based on the hydrological modeling performance on the 350 gauged catchments, as represented by the KGE calibration score, the MSWEP precipitation and ERA5 temperature datasets were*

*found to be the best combination used in computing the streamflow for the 795 ungauged catchments."*

[Page 8, line 213] I would have liked to read a bit more about the variance analysis, about the methods and the aim of this analysis. E.g. What are the variance components and what do they tell you? Is such an analysis computationally expensive to run?

We will add the main relevant details in the revised version.  Essentially, for each catchment we get 360 values for each metric, each value related to a unique combination of 1 GCM, 1 precipitation dataset, 1 hydrology model and 1 temperature dataset.  The variance analysis attributes the percentage of the total variance of this vector of 360 values, to these components, including the interactions between these components, interactions meaning that the behaviour of one source depends on another source (for example, precipitation dataset may generate lots of variance with some GCM but not for others).  Computing the main effect and first order interactions is relatively cheap, computationally speaking, but higher orders (which typically carry much less variance) become exponentially costlier.

*"An n-dimensional analysis of variance (ANOVA-N) was used to quantify the contribution of the different uncertainty sources to the overall variance (Von Storch and Zwiers, 2001). This method has been applied in many previous studies for this purpose (Addor et al., 2014; Bosshard et al., 2013; Trudel et al., 2017). For each catchment, 360 values for each metric are obtained, each related to a unique combination of 1 GCM, 1 precipitation dataset, 1 hydrology model and 1 temperature dataset. The variance analysis attributes the percentage of the total variance of this vector of 360 values, separating the main effects (the independent contribution of each of the 4 components, and the interactions between these components. The interactions imply that the behaviour of one source depends on another source (for example, precipitation dataset may generate lots of variance with some GCM but not for others). Computing the main effect and first order interactions is relatively cheap, computationally speaking, but higher orders (which typically carry much less variance) become exponentially costlier. For the four uncertainty components under study (GCMs, precipitation and temperature datasets, and hydrological models), a total of 11 variance components can therefore be computed: 4 main effect components as well as 6 first-order, 3 second-order, and 1 third-order interaction components."*

[Page 8, lines 214-216] Could you please provide an overview of the metrics computed, perhaps in a table?

 Good idea. We defined the computed streamflow metrics in the revised version.

 *"The analysis of variance was performed for six streamflow metrics out of the fifty-one metrics defined in Arsenault et al. (2020) for each of the 1145 catchments. These six metrics cover a wide range of streamflow conditions: mean annual (Mean Q), seasonal (Winter Q and Summer Q) values, the 5th and 95th distribution quantiles (QQ5 and QQ95, respectively), as well as annual daily extreme (QX1) metrics."*

[Page 8, lines 232-233] Please clarify that this observation is with regards to calibration for these 350 catchments. It could otherwise be misinterpreted taken out of context.

Will do. Your interpretation is correct.

*"Based on the models calibration performance, both temperature datasets perform very similarly across all combinations, with ERA5 generally slightly outperforming CPC. Figure 3 clearly shows that most of the variability seen originates from the precipitation datasets. Four precipitation datasets are ahead of the field. They are, in order of performance: the merged product MSWEP, followed by the two satellite datasets; CHIRPS and PERSIANN, and the ERA5 reanalysis dataset. The gauge-based precipitation datasets (e.g., GPCC and CPC), and the ERA-I reanalysis follow with a similar performance. Finally, the CFSR and JRA55 reanalysis are the worst-performing products for hydrological model calibration."*

[Page 9, line 256] The mean Summer Q also appears to show a different signal from the other metrics.

It is somewhat in the middle between the low-flow and the other metrics. This is not entirely surprising, since summer is a dry season over East Africa (between the spring and autumn monsoons) and therefore somewhat related to the low-flow metric. We will make a remark to this effect in the revised version.

*"As was shown in Table 3, the low-flow metric displays a pattern that is much different from the other five metrics, with HM being important and GCM, being the lowest. Also, the HM and P-HM show significant contribution to the uncertainty in the Summer Q metric."*

[Page 9, lines 260-261] This is arguable and quite complicated to see. Perhaps putting the metrics in columns and the contributors in rows might help see these better?

This is a complex figure. We're not sure that doing so will help but we will try it and judge if one is preferable to the other.

[Page 9, lines 268-269] Which variance contributor is Fig. 6 shown for?

We will clearly have to be more explicit about this Figure in the revised version. The analysis of variance shows the percentage of total variance for each contributor, irrelevant to the total absolute variance. Trying to attribute relative variance to something that has little variance to begin with may not be that useful. So Figure 6 shows the absolute variance to try to make sense of this.

*"In other words, a variance analysis of a metric with very little absolute variance could be misleading. Consequently, Figure 6 displays the standard deviation of the 360 streamflow values computed for each streamflow metric and for each watershed. Therefore, Figure 6 does not represent the variance contribution of any given component of the hydroclimatic chain, but represents the total variance of all components combined. A low value indicates that a streamflow metric shows little variability across its 360 values. This would be expected for example for catchments with a high aridity index resulting in very transient flow."*

[Page 10, lines 284-285] It seems to be that most of the drop is seen between ensembles 1 and 2, rather than 2 and 3. Please also add the ensemble numbers 1-4 from table 4 to Fig. 7.

We see sizable drops from 1 to 2 as well as from 2 to 3. We will add the ensemble numbers to the all relevant Figures in the revised version.

[Page 10, lines 290-292] Hydrological model uncertainty appears dominant over precipitation uncertainty for low flows.

Correct. We will state it clearly in the revised version.

*"Results outline that GCM uncertainty is the dominant source of uncertainty when using the reduced precipitation ensemble, with the exception of the low-flow metric, for which hydrological model uncertainty is dominant."*

[Figure 2] In order to be clearer for the readers, please consider adding clear subheading for each box in this diagram.

This is a good suggestion. We will implement it.

**RC2 - Anonymous Referee #2 comments:**

The manuscript by Tarek et al compares nine precipitation and two temperature datasets for 1145 African catchments to understand the uncertainties they introduce when used in hydrological modelling applications. The study is substantial, addresses a significant community challenge, and offers practical guidance to others in the field. On that basis, I judge that it will be of clear interest to the readership of HESS, with substantial impact in Africa and insights applicable more widely. There are three major points, which the authors should address in revision, and some minor points which they might wish to consider too.

Thank you for your comments. Please see the point-by-point responses below.

**Key comments**

-         Bias correction is applied but only five lines are given to the description of the method used. It is essential to have more information on the bias corrections made, especially at rarely seen precipitation quantiles, given the sparsity of data in this region.

In the original manuscript, we tried to balance concision and provide enough methodological details. We will gladly add additional information on the bias correction method and especially its treatment of the larger quantiles. The devil is in the details when it comes to bias correction methods, and this is particularly the case for the large quantiles. We will therefore provide an extended description of the MBCn method of Cannon (2018).

*"Most climate change impact studies have been applying univariate bias correction methods to correct climate model outputs. Univariate approaches cannot account for the temporal dependence between precipitation and temperature (and other variables). For example, if a model has a cold temperature bias and a dry precipitation bias, these biases would be corrected individually, whereas in reality precipitation and temperature are correlated (e.g. Wu et al., 2013). Multivariate techniques have been introduced as an alternative to overcome this deficiency. In this study, the N-dimensional multivariate bias correction algorithm (MBCn) was used (Cannon, 2018).*

*MBCn is an image processing technique extension that transfers all statistical characteristics between the historical and projected periods while preserving the change projected for all quantiles of the distribution. The algorithm consists of three main steps: (1) application of an orthogonal rotation to both model and observational data; (2) correction of the marginal distributions of the rotated model data using quantile mapping, and (3) application of an inverse rotation to the results. These three steps are repeated until the model distribution matches the observational distribution. This computational complexity is one disadvantage of that method, as it requires several iterations to correct the projected outputs. However, MBCn is arguably the best-performing quantile-based method available (Adeyeri et al., 2020; Meyer et al., 2019). "*

-         Hydrological modelling uncertainty (L240): more information is needed to attribute uncertainty in the hydrological modelling part of the work. It is unclear at present which components of the hydrological models contribute greatest uncertainty, especially with the HMETS model.

This is a good point, albeit a difficult one to answer since this is a topic that has not been studied until very recently. The impact of the hydrological model structure on the uncertainty is now relatively well established in the scientific literature, but the main sources of uncertainty (e.g. PET, vertical and horizontal water movement, snow model) have not been. It is not feasible to do so in the context of this work since it would require us to redo the entire analysis by decoupling the hydrology model into separate parts and recalculating all variance components. This would be a formidable task indeed. However, in light of recently published work (e,g, Dallaire et al., 2020; Duethmann et al., 2020; Van Kempen et al,.2020), we propose to enhance the discussion to offer insights on what parts of the hydrological model structure are likely critical to climate

change impact studies, and which part are likely to carry most of the uncertainty. We will also emphasize this point as a key area of future research.

Dallaire et al., 2020, Uncertainty of potential evapotranspiration modelling in climate change impact studies on low flows in North America .  Hydrological Sciences Journal, In press.

Duethmann, D., Blöschl, G., & Parajka, J. (2020). Why does a conceptual hydrological model fail to correctly predict discharge changes in response to climate change?. Hydrology and Earth System Sciences, 24(7), 3493-3511.

Van Kempen, Gijs, Van Der Wiel, Karin, and Melsen, Lieke Anna. The impact of hydrological model structure on the simulation of extreme runoff events. Natural Hazards and Earth System Sciences Discussions, 2020, p. 1-24.

*"A better understanding of how hydrological model components affect uncertainty would therefore be very valuable for climate change impact studies (e.g. Dallaire et al., 2021;* Duethmann *et al., 2020; van Kempen et al., 2021). Taking the above into consideration, it is therefore likely that the contribution of hydrological models is underestimated here. The number of components in a variance attribution study is an important issue. However, the contribution to variance is related to how dissimilar the ensemble members are and not strictly to its numbers. As such, the ensemble with the fewest members can still provide the largest contribution to variance. There is therefore no need for all ensembles to have the same number of members, but rather to have enough credible members to cover the uncertainty. Despite having only two temperature datasets here, adding more temperature datasets is unlikely to change the results considering how little uncertainty is present in the two datasets when compared to other sources. Temperature is the easiest variable to measure and to extrapolate, especially when compared to precipitation. It is therefore expected that precipitation uncertainty would normally dwarf the contribution of temperature. Based on previously published work, 10 GCMs is very likely more than enough to frame the uncertainty contribution from this source (e.g. Wang et al., 2020)."*

-    Applicabilty (L360 onwards): one of the key virtues claimed in the introduction is the guidance offered to future researchers tackling this problem afresh. Yet in this section the reader is left with the sense that there is no clear advantage to any particular dataset, with conclusions left as context dependent. Clearly that's a valid finding but it needs to be reflected in the stance that the paper offers in its opening and concluding paragraphs.

We agree that we probably should have been more committed in our conclusions. Some precipitation datasets are clearly better than others and we will spell it out more clearly in the revised version. While there will always be some 'context-dependent' issues, by large, we will provide better guidance in the revised version.

*"Some level of guidance for impact modelers can nonetheless be offered from the results of this work. Precipitation is the key driver of dataset uncertainty and should therefore be evaluated in climate change studies alongside the more traditional sources of uncertainty. In cases where it is*

*not possible to select multiple precipitation datasets, the results presented in Figure 3 and in Tarek et al. (2020a) indicate that MSWEP merged product dataset is the best performing one, with CHIRPS and ERA5 being the next best. The gauged-only based products were clearly not the best-performing ones over Africa in contrast to a similar study performed over North-America (Tarek et al., 2020b). This performance ranking is however only based on the KGE calibration metric. While the KGE is a good overall performance metric, it is possible that using a different performance metric might affect this ranking. Streamflow data also come with many potential quality issues that must be taken into consideration (e.g. Tomkins, 2014; Hamilton and Moore, 2012). However, in the overwhelming majority of cases, there are no competing streamflow datasets from which to study uncertainty from. But flawed streamflow records will impact hydrological model calibration and performance, and may therefore indirectly contribute to hydrological model uncertainty."*

**Minor comments**

Reference formatting is inconsistent with many additional brackets, etc.

This will be taken care of in the revised version.

[Page 6, line 173] hypodermic -> subsurface.

This will be taken care of in the revised version.

*"It has four reservoirs (surface runoff, subsurface flow from the vadose zone reservoir, delayed runoff from infiltration and groundwater flow from the phreatic zone reservoir)."*

Figure 5: annotations and labels are hard to read; please enlarge

We will redraw the Figures to make the text easier to read.

**SC1 - Short Comment by Yves Tramblay:**

I found this study really informative about the issues related to the choice of "observed" rainfall for hydrological impact studies of climate change. However, the regionalization approach considered here is barely described in section 3.2, when the results for 795 stations out of 1145 rely on this regionalization procedure. I am not aware of other studies attempting to regionalize the model parameters of the GR4J or HMETS models at the scale of Africa, so the results of this regionalization procedure surely deserve more than one sentence (line 202). In fact, I really believe this type of regional analysis would require a study on its own.

Point well taken. We will add additional information on the regionalization process. We agree that this issue is worth a separate study and we just very recently submitted a paper dedicated to the

regionalization aspect of this work. We will summarize the results of the second paper in the revised version of this one to get a better sense of the regionalization method performance.

*"This regionalization study is one of the very few performed over Africa and will be detailed in another paper. It showed that the best regionalization methods were consistent with the ones identified in other regions of the world, and that regionalization performance was similar to that obtained in studies elsewhere around the world."*

What is the efficiency of the regionalization methods tested in a cross-validation framework? Beside the spatial proximity, how are the "physical similarity" and the "multiple-linear regression" methods implemented? What are the predictors, since the authors only mention watershed delineation in the manuscript?

This is a good point, which is in line with the previous comment regarding the necessity of having a separate paper to go into this level of detail. The main results will be summarized in the revised version of this paper. To answer the question more specifically, the multiple linear regression method did not work very well, but the physical similarity performed similarly to (and in some cases better than) the spatial proximity method. These 2 "catchment-descriptor-based" methods used land cover properties (% grassland, % forest, etc.), mean annual rainfall, aridity index, mean slope and other such properties to identify the relationships between parameters and descriptors (or to find the most similar donors).

Another aspect is the presence of dams and reservoirs. Many African rivers are regulated and no mention is given in the data section 2.2.3 if the selected rivers are regulated or not. We recently released a large dataset or river discharge in Africa (https://doi.org/10.23708/LXGXQ9) and from the metadata it can be seen that about one third of the basins are regulated. It could explain the bad modelling results for some basins, since the hydrological models are not validated against independent data in the present study (line 182). It is likely that the modelling results with different satellite products can be impacted by river regulation in some basins.

This is a good point and one that we have chosen not to investigate in the paper. We have KGE values exceeding 0.6 for 75% of the catchments for the best-performing hydrological model which we took as proof of the absence of any major regulation. Most of the catchments with a lower KGE performance are in arid and semi-arid regions, where hydrological modeling is more challenging, and especially so for simple models like the ones used in this study. But clearly, it is certainly possible that the lower performance over some of the catchments is due to regulation. We could have done some homogeneity testing to try to detect changes that could be related to flow regulation, but attributing inhomogeneity to a specific cause is not always simple. Considering the optic of the paper, we don't think we need to start investigating this issue in details, but the potential issue of flow regulation should definitely be mentioned in the discussion. We will add a paragraph or two on this issue. Thanks for providing the references below, which we will also add to support the problem of data scarcity over Africa.

Tramblay, Y., Rouché, N., Paturel, J. E., Mahé, G., Boyer, J. F., Amoussou, E., ... & Lachassagne, P. (2020). The African Database of Hydrometric Indices (ADHI). Earth System Science Data Discussions, 1-21.

*"The streamflow records in the GRDC database have all undergone a quality control process, but there is always the possibility that some level of regulation may affect the data (Tramblay et al., 2020). No direct homogeneity testing was performed to detect potential changes due to regulation, but an indirect quality assessment was done through the hydrological modeling performance during the calibration process."*

Finally, I am a bit surprised by the figures 5, 6 and 8, you have river runoff over the Sahara Desert?

This is a good point that should have been mentioned. This will be done in the revised version. There is indeed little to no runoff on most parts of the Sahara desert, a consequence of little to no precipitation over most regions. However, it does rain (and even snows in some very rare cases). Predicted runoff was highly intermittent but consistent with precipitation datasets. We checked satellite imagery for many catchments and all images showed some drainage patterns (unconnected at the regional scale) consistent with very sporadic rainfall. Figures 5 and 8 show the relative contribution to variance of each uncertainty source. The absolute variance is however extremely small over this region, so these figures should be interpreted with care over the Sahara.

*"Above 20°N, there is generally less than 100 mm of total annual precipitation, and some level of care should therefore be taken when analyzing results in relative contribution to variance. The relative contribution to variance is not related to absolute mean streamflow values and therefore the colour scale is the same for major rivers and smaller intermittent streams. Many of the catchments above 20°N run dry for a large part of the year."*